# Learning Strategic Language Agents in the Werewolf Game with Iterative Latent Space Policy Optimization

Zelai Xu [1]   Wanjun Gu [1]   Chao Yu [1 2]   Yi Wu [* 1 3]   Yu Wang [* 1]

## Abstract

Large language model (LLM) agents have recently demonstrated impressive capabilities in various domains like open-ended conversation and multi-step decision-making. However, it remains challenging for these agents to solve strategic language games, such as Werewolf, which demand both strategic decision-making and free-form language interactions. Existing LLM agents often suffer from intrinsic bias in their action distributions and limited exploration of the unbounded text action space, resulting in suboptimal performance. To address these challenges, we propose Latent Space Policy Optimization (LSPO), an iterative framework that combines game-theoretic methods with LLM fine-tuning to build strategic language agents. LSPO leverages the observation that while the language space is combinatorially large, the underlying strategy space is relatively compact. We first map free-form utterances into a finite latent strategy space, yielding an abstracted extensive-form game. Then we apply game-theoretic methods like Counterfactual Regret Minimization (CFR) to optimize the policy in the latent space. Finally, we fine-tune the LLM via Direct Preference Optimization (DPO) to align with the learned policy. By iteratively alternating between these steps, our LSPO agents progressively enhance both strategic reasoning and language communication. Experiment on the Werewolf game shows that our agents iteratively expand the strategy space with improving performance and outperform existing Werewolf agents, underscoring their effectiveness in free-form language games with strategic interactions.

[*]Equal advising  [1]Tsinghua University, Beijing, China [2]Beijing Zhongguancun Academy, Beijing, China [3]Shanghai Qi Zhi Institute, Shanghai, China. Correspondence to: Chao Yu <yuchao@tsinghua.edu>, Yi Wu <jxwuyi@gmail.com>, Yu Wang <yu-wang@tsinghua.edu>.

*Proceedings of the 42$^{nd}$ International Conference on Machine Learning*, Vancouver, Canada. PMLR 267, 2025. Copyright 2025 by the author(s).

## 1. Introduction

Developing intelligent agents that can reason rationally, make strategic decisions, and interact with humans has been a long-term goal in artificial intelligence (AI) research (Wooldridge & Jennings, 1995; Russell & Norvig, 2016). In recent years, large language model (LLM)-based agents have made significant strides towards this goal by exhibiting strong performance in open-ended conversation and multi-step decision-making (Brown et al., 2020; Ouyang et al., 2022). Trained on massive text corpora, LLM-based agents have demonstrated remarkable versatility across various domains, ranging from web navigation (Nakano et al., 2021; Yao et al., 2022b) and code generation (Chen et al., 2021; Yang et al., 2024) to video game environment (Wang et al., 2023a) and real-world scenarios (Ahn et al., 2022; Brohan et al., 2023). Beyond single-agent tasks, LLM-based agents have also shown potential in multi-agent interactions including collaborative teamwork (Li et al., 2023), adversarial gameplay (Meta et al., 2022), and human-AI interation (Park et al., 2023; Liu et al., 2023).

Among these interactive domains, strategic language games such as Werewolf present unique challenges because they require both high-level strategic decision-making and free-form conversational abilities. Unlike classic games with predefined and limited actions, such as board games (Silver et al., 2016; 2018), card games (Moravčík et al., 2017; Brown & Sandholm, 2018), and video games (Mnih, 2013; Vinyals et al., 2019), Werewolf relies heavily on free-form conversation to achieve agreements and perform strategic deceptions. Players must communicate, bluff, and infer hidden roles through unrestricted, natural language interactions. This free-form language space expands the strategic possibilities and introduces additional complexity unmatched by more rigidly defined domains. As a result, Werewolf serves as an ideal environment for developing strategic agents with language-grounded decision-making capabilities.

However, developing a strategic language agent that can interact with humans in Werewolf or other free-form language environments is still challenging. Classic game-theoretic methods like Counterfactual Regret Minimization (CFR) and reinforcement learning (RL) have proven successful in games like Go and Poker, thanks to their ability to handle

finite action spaces. Yet Werewolf has a free-form action space, making direct application of these methods computationally infeasible. Mapping every possible utterance to an action in the original text space becomes prohibitively large, leading to immense difficulty in strategy representation and equilibrium-finding. An alternative approach is to build language agents with LLMs. These methods typically rely on prompt engineering without training the base LLM, which means their success depends entirely on the general reasoning capabilities of LLMs to generate actions. Unfortunately, prompt-based methods suffer from intrinsic bias in their generated actions (Xu et al., 2023c), resulting in suboptimal performance in strategic language games like Werewolf. Moreover, these agents exhibit limited exploration of novel strategies because they fully rely on the LLMs to generate actions, making the agents constrained by the capability of the base LLMs. Some work (Chen et al., 2023a; Wu et al., 2024) mitigates these issues by fine-tuning the LLM for a specific task, which requires expensive human labor for high-quality data. Xu et al. (2023c) partially tackles the bias issue by additionally training a small network to calibrate the LLM output distribution. However, it still relies on a fixed LLM to produce action candidates, leaving the exploration issue unaddressed. This raises an important question: *Can we have a method that leverages both the specialized reasoning capabilities for decision-making and the generalization capabilities of LLMs?*

In this work, we propose an iterative Latent Space Policy Optimization (LSPO) framework to build strategic language agents for free-form language games, taking Werewolf as our testbed. Our approach combines structured game-theoretic methods with language models by introducing a discrete latent strategy space. Our method consists of three components. We first map the free-form utterances into a manageable, discrete strategy space to yield an abstracted game. Then we apply game-theoretic methods like CFR to learn the optimal policy in the latent space. Finally, we fine-tune the LLM via Direct Preference Optimization (DPO) (Rafailov et al., 2024) to align with the learned policy and expand the latent space. By iterating between these latent space CFR steps and LLM fine-tuning, our method yields an evolving agent that addresses both the intrinsic bias issue with game-theoretic methods and the action exploration issue with latent space expansion, leading to strong performance in the Werewolf game.

We perform extensive experiments in the Werewolf game to demonstrate the effectiveness of our LSPO framework. We first analyze how the latent strategy space evolves between iterations to show that our agents learn increasingly complex and strategic behaviors. Then we quantitatively evaluate the prediction accuracy and win rate of our LSPO agent to show the improving performance with respect to iterations. Next, we compare our agents against state-of-the-art Werewolf

agents and find that the LSPO agent achieves the highest win rate. We also conduct ablation studies to assess the effectiveness of our design in the LSPO framework.

## 2. The Werewolf Game

Werewolf is a popular social deduction game where players with hidden roles cooperate and compete with others in natural languages. The Werewolf side needs to conceal their identities and eliminate the other players, while the Village side needs to identify their teammates and vote out the Werewolves. Players are required to have both language proficiency for communication and strategic ability for decision-making to achieve strong performance in the Werewolf game. We consider a seven-player game with two Werewolves being the Werewolf side and one Seer, one Doctor, and three Villagers being the Village side. Detailed descriptions of the game's rule, observation space, and reward function can be found in Appendix A.

### 2.1. Game Environment

We consider a text-based seven-player Werewolf game that proceeds through natural languages. We exclude other information like the speaking tone, facial expression, and body language (Lai et al., 2022). This pure text-based environment is a common setup in the literature (Xu et al., 2023a;c; Wu et al., 2024; Bailis et al., 2024).

**Roles and Objectives.** At the beginning of each game, the seven players are randomly partitioned into two sides. The Werewolf side has two Werewolf players who know each other's role and aim to eliminate the other players while avoiding being discovered. The Village side has one Seer who can check the role of one player each night, one Doctor who can protect one player each night, and three Villagers without any ability. The players in the Village side only know their own role and need to share information to identify the Werewolves and vote them out.

**Game Progression.** The game proceeds by alternating between night round and day round. In the night round, players can perform secret actions that are only observable by themselves. More specifically, the two Werewolves can choose a target player to eliminate, the Seer can choose a target player to investigate whether the player's role is Werewolf, and the Doctor can choose a target player to protect the player from being eliminated. The Doctor does not know the target player chosen by the Werewolves. If the Doctor chooses the same target player as the Werewolves, then no player is eliminated in this night round, otherwise, the Doctor fails to protect any player, and the target chosen by the Werewolves is eliminated.

**Observations and Actions.** The language observation of each agent is a list of natural languages that log the game

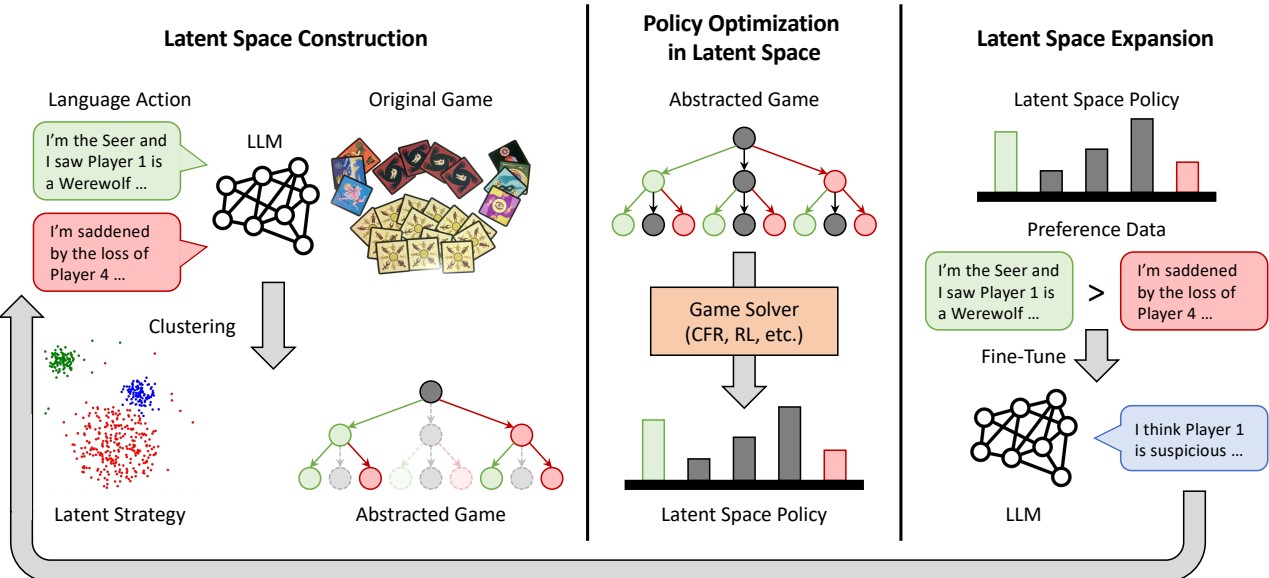

Figure 1: Overview of the Latent Space Policy Optimization (LSPO) framework. Each iteration consists of three components. (1) Latent space construction: generate language actions with the LLM and cluster the vast language action into a finite latent strategy space. (2) Policy optimization in latent space: reformulate the original game as an abstracted game and apply game-theoretic methods to learn latent space policy. (3) Latent space expansion: fine-tune the LLM to align with the latent space policy and generate new strategies to expand the latent strategy space.

history to the current step. This list include both private information that are only observable to the current player and public information that are shared by all players. The private information includes the role of the current player, the secret actions in the night round for the Werewolf, Seer, and Doctor, and the teammate for the Werewolf. The public information includes the ID of the current player, the eliminated player in each night and day round, the discussion, and the voting result in each day round.

Player actions are also in the form of natural language and can be categorized into three types: secret actions, which are secret actions performed during the night, such as choosing a target player to eliminate, investigate, or protect; discussion actions, which are statements made during the day to influence other players' perceptions and decisions; and voting actions, which are choices made during the voting round to vote for on player or choose not to vote.

### 2.2. Challenges for Language Agents

Unlike board, card, or video games with a finite set of pre-defined actions, Werewolf has a free-form language action space. The vast space of natural language actions poses two key challenges for language agents to achieve strong performance in the Werewolf game.

**Intrinsic Bias in Action Generation.** As observed in simple games like Rock-Paper-Scissor (Xu et al., 2023c), pure

LLM-based agents tend to exhibit intrinsic bias in their action generation, which is inherited from the model's pre-training data. This issue is more pronounced in complex language games like Werewolf, where the opponents can exploit these predictable biases to counteract the agent's move. Therefore, mitigating intrinsic bias is essential for language agents to reduce exploitation and achieve strong performance in strategic language games.

**Exploration of Unbounded Action Space.** Due to the immense combinatorial space induced by free-form text, it is impractical to map every possible utterance to an action in the language space. On the other hand, manually engineering or prompting an LLM to produce a limited set of actions may fail to capture the full strategic landscape. Even if an agent optimally masters the action distribution within a limited subset, it could be easily exploited by out-fo-distribution utterance. Consequently, inadequate exploration of the action space could result in suboptimal performance in free-form language games like Werewolf.

## 3. Latent Space Policy Optimization

To tackle the intrinsic bias and the exploration issue, we propose an iterative Latent Space Policy Optimization (LSPO) framework. Our method combines game-theoretic optimization with LLM fine-tuning and operates on an expanding latent strategy space to iteratively improve the agent's

decision-making ability and action exploration. As shown in Figure 1, our framework has three components including latent space construction, policy optimization in latent space, and latent space expansion. More implementation details can be found in Appendix C.

## 3.1. Latent Space Construction

One of the key challenges in free-form language games like Werewolf is achieving broad exploration of the unbounded text space while maintaining a computationally tractable action representation for game-theoretic methods. To strike a balance between exploration and tractability, we propose to abstract the vast language action space into a finite set of latent strategies, which we then expand over iterations for better exploration. Specifically, our latent space construction in each iteration involves two steps including latent strategy generation and clustering.

**Latent Strategy Generation.** In our setting, secret actions and voting actions are already discrete and therefore do not require further abstraction. We focus instead on the free-form discussion actions, which we aim to capture as latent strategies. We assume that each role in the game has the same set of latent strategies across all discussion rounds and collect the latent strategies for each role by letting the current LLM agent self-play as different roles for multiple trajectories. To further improve the exploration of latent strategies, we prompt the LLM to generate $N$ strategically distinct discussion candidates and randomly choose one to execute in the game. This process encourages diversity in the collected discussion actions and generate a set of latent strategies in natural language for each role.

**Latent Strategy Clustering.** Although we generate a set of latent strategies for each role, they are still in the form of natural language. To transform them into a discrete latent strategy space, we embed each discussion action into a vector representation using an embedding model such as "text-embedding-3-small" that captures its semantic and contextual information. We then apply a simple $k$-means clustering algorithm to partition the embedded utterances into $k$ clusters, where each cluster represents a distinct latent strategy. Clustering reduces the infinite free-form text space to a finite set of abstract strategies, paving the way for subsequent game-theoretic optimization. By interpreting each cluster as a latent action, we can more efficiently search for and optimize strategic policies with minimal sacrifice of exploration of language space.

## 3.2. Policy Optimization in Latent Space

Another challenge in free-form language games is to address the intrinsic bias in the agent's action distribution. After constructing a discrete latent strategy space, we can reformulate the original game with unbounded language space as an abstracted game with a finite latent strategy space. This reformulation allows us to apply standard game-solving techniques such as Counterfactual Regret Minimization (CFR) or reinforcement learning (RL) methods to learn near-optimal strategies that overcome the intrinsic bias. In our implementation, we employ CFR as the game solver.

**Abstracted Game Formulation.** To represent the game in a compact, finite form, we replace the free-form discussion actions with the discrete latent strategies from latent space construction. Specifically, the abstracted game is formalized as an extensive-form game (EFG), where the secret action and voting action remain the same, and the discussion action is replaced by the latent strategy. The state in the abstracted game is a vector including information like the player's role, secret action, etc., and history of past latent strategies. The transition dynamics and payoff function remain unchanged in the abstracted game. This representation retains the key strategic elements of the original game while reducing the complexity of the action space, making large-scale game-solving computationally tractable. Detailed description of the abstracted game can be found in Appendix A.

**Policy Optimization.** Once the game is represented in this discrete form, we apply CFR to learn a policy and solve the abstracted game. Classical CFR (Zinkevich et al., 2007) iteratively improves policies by minimizing counterfactual regret $R$ for each information set. For each iteration $t$, the regret for each action $a$ in the latent space is updated by:

$$R_t(a) = R_{t-1}(a) + u(\sigma_t^a, \sigma_t^{-a}) - u(\sigma_t), \qquad (1)$$

where $u(\sigma_t^a, \sigma_t^{-a})$ is the utility of taking action $a$ under the current strategy profile $\sigma_t$, and $u(\sigma_t)$ is the utility under the full strategy profile. We use neural networks to approximate regret value to scale CFR to more complex games and learn a policy for each different role in the Werewolf game. By repeatedly simulating self-play among agents employing Deep CFR in the abstracted game, each role's policy converges to a near-optimal strategy profile. The resulting latent space policies address the intrinsic bias in action distribution and achieve strong strategic play in the abstracted game.

## 3.3. Latent Space Expansion

To further improve the agent's performance in free-form language games, the latent space must remain sufficiently expressive to cover novel strategies and resist exploitation by out-of-distribution actions. We achieve this by fine-tuning the LLM to align with the learned policy in the abstracted game and then re-generating and expanding the latent strategy space using the fine-tuned LLM. This iterative process progressively increases exploration of the action space, enabling stronger and more robust decision-making.

**Alignment to Latent Space Policy.** We employ Direct Preference Optimization (DPO) (Rafailov et al., 2024) to

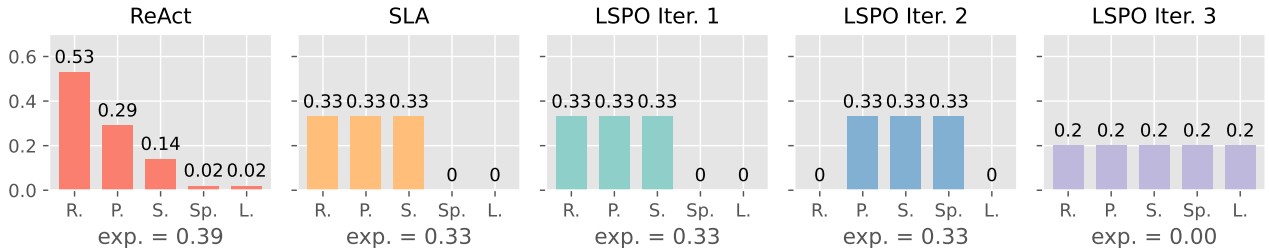

Figure 2: The action distributions and exploitabilities (exp.) of different agents in the *Rock-Paper-Scissors-Spock-Lizard* game. The Nash equilibrium is to choose each action with an equal probability of $1/5$ and has an exploitability of $0$.

fine-tune the LLM so that its open-ended language outputs align with the near-optimal strategies derived from the abstracted game. To construct the preference dataset required by DPO, we leverage game trajectories generated during latent space construction. We record the language observation for the LLM agent at each discussion phase as the prompt, and use the $N$ discussion candidates as the response candidates. Each of the discussion candidates can be mapped to one of the latent strategies, and the preference label is determined by the regret value of the latent strategies. Intuitively, a discussion action with a lower regret value is preferred. With this preference dataset, we perform DPO to align the LLM toward the learned policy in the abstracted game for better performance in the original game.

**Update of Latent Space.** Once the LLM is fine-tuned, it can produce a broader distribution of actions that reflect the refined policy. We exploit this enhanced generative capacity to expand the latent space in the next iteration. Specifically, we repeat the latent strategy generation and clustering procedures with the fine-tuned LLM to re-generate and expand the latent strategy space. This updated latent space offers increased exploration of potential strategies, enabling subsequent policy optimization to discover previously unexplored high-reward actions. Through iterative alignment and expansion, the agent continually refines its discussion strategies and achieves strong play in the free-form language game.

## 4. Experiments

To demonstrate the effectiveness of the LSPO framework, we first consider a proof-of-concept game to show how LSPO overcomes intrinsic bias and addresses the exploration issue. Then we conduct extensive experiments in the Werewolf game with `Llama-3-8B-Instruct` as our base model. We visualize how the latent strategy space evolves to show that our agents progressively acquire more complex strategic behaviors. We then quantitatively evaluate the performance of our LSPO agent using prediction accuracy and win rate to show the improving performance over iterations. We also compare the LSPO agent with four state-of-the-art

agents, showing that our agents achieve the highest win rate on both the Werewolf side and the Village side. We further perform ablation studies to assess the effectiveness of specific designs in our framework. More implementation and experiment details can be found in Appendix B.

### 4.1. Proof-of-Concept Example

We consider *Rock-Paper-Scissors-Spock-Lizard*, a five-choice extension of the classic *Rock-Paper-Scissors* game. Although it is not a free-form language game, it serves as a simple proof-of-concept game that highlight the motivation of our method. The intrinsic bias in action distributions is inherited from the imbalanced LLM pre-training data, and the exploration issue is introduced by the two additional actions of Spock and Lizard. The Nash equilibrium (NE) of this game is to choose each action with an equal probability of $1/5$. We compare LSPO agents of different iterations with two baselines, including Reason and Act (ReAct) (Yao et al., 2022b) and Strategic Language Agent (SLA) (Xu et al., 2023c), and the action distributions and exploitabilities of different agents are shown in Figure 2.

The evaluation results show that the LSPO agent of iteration 3 learns the NE of the game while other agents fail. ReAct agent suffers from the intrinsic bias issue and has higher probabilities to choose Rock, Paper, and Scissors, and much lower probabilities to choose Spock and Lizard. SLA, on the other hand, is hindered by inadequate action exploration. SLA uses LLMs to propose $N$ actions and RL to learn the optimal policy. A typical value of $N = 3$ results in a subgame without the other 2 action, and the NE of the subgame is not the NE of the original game. Our LSPO agent addresses these two challenges by iteratively expanding the action space. As it covers the full action space with 3 iterations, the LSPO agent learns the NE of the game.

### 4.2. Latent Space Visualization

To gain insight into how LSPO organizes free-form language actions into discrete latent strategies, we first visualize the latent strategy space constructed at different training itera-

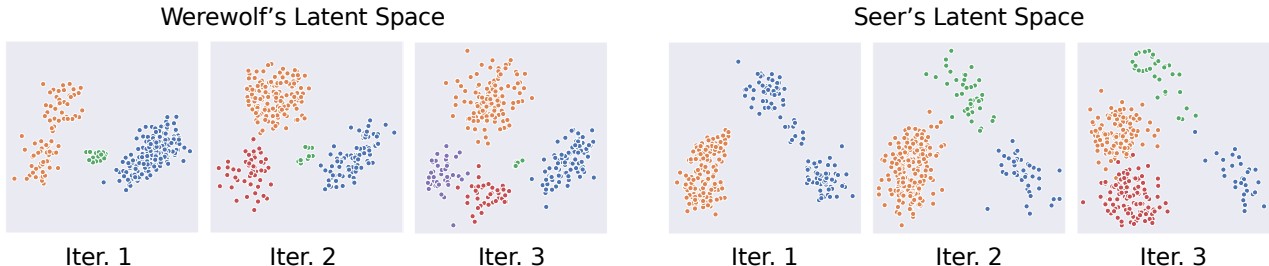

Figure 3: Visualization of the latent strategic space of Werewolf and Seer in different LSPO iterations.

tions. Specifically, for each role in the Werewolf game, we gather the utterances generated by the LSPO agent in 100 games, embed them with the sentence encoder, and apply dimensionality reduction for projection. The visualization of latent spaces for the Werewolf and the Seer in different iterations is shown in Figure 3. Earlier iterations yield relatively indistinct clusters, reflecting a lack of strategic diversity. Over successive iterations, clearer and more refined clusters emerge, indicating that the LSPO agent evolves toward an increasingly structured latent space and learn to express different strategic intentions such as accusing specific roles, defending teammates, and bluffing.

**Werewolf's Latent Space.** In the first iteration, the latent space of the Werewolf is dominated by three main clusters. The blue cluster corresponds to a simple strategy of concealing its role or pretending to be a villager, while the smaller orange cluster reflects strategies like pretending to be a Seer or a Doctor. There is even a green cluster corresponding to unintentionally revealing the true role of a Werewolf, which is obviously a flawed strategy. As training proceeds, we see more sophisticated patterns emerge. The flawed strategy of disclosing one's Werewolf role disappears, and the agent begins to incorporate deliberate bluffs and misdirections instead. For example, the red cluster features the agent pretending to be a Seer and providing fabricated investigative results to sow confusion, and the purple cluster centers on defending the teammate and redirecting suspicion onto other players, leveraging more nuanced language and reasoning to guide the conversation toward scapegoats. This refined partitioning demonstrates that the Werewolf agent progressively covers an increasing number of latent strategies.

**Seer's Latent Space.** In the first iteration, the Seer's latent space is relatively coarse, containing primarily two strategies including staying silent about its true role or revealing its role and sharing information. This shows a limited range of strategic diversity in the early stage. As training proceeds through the second and third iterations, the Seer's latent space becomes more diverse. The emergent red cluster features direct accusations once the Seer identifies a Werewolf, while the green cluster corresponds to concealing the role

yet subtly guiding discussions to protect verified teammates. Notably, by the final iteration, the model develops a voting coordination strategy in which the Seer explicitly asks all the Villagers to vote for a strongly suspected Werewolf to maximize the Villager's chance of success. This progression implies that the Seer agent increasingly learns to balance openness and secrecy, aligning its communication style with the evolving game context to better support the Village side.

### 4.3. Iterative Performance Evaluation

We then evaluate how the performance of our LSPO agent progresses with more iterations, demonstrating that our framework produces increasingly stronger strategic language agents over time. We focus on two key metrics including prediction accuracy and win rate.

**Prediction Accuracy.** Accurate role identification is a critical aspect of Werewolf, as it underpins effective decision-making and voting. Therefore, we measure the agent's ability to predict the roles of other players with an additional prediction phase before each voting phase in a Werewolf game. Specifically, we use the final-iteration LSPO agent as the fixed opponent and let LSPO agents at different iterations play against this opponent for 100 games. For the Werewolf side, a higher prediction accuracy of crucial roles like Seer and Doctor allows them to eliminate these threats earlier. Conversely, for the Village side, a higher prediction accuracy of Werewolves improves their chance to vote out the Werewolf and win the game.

**Win Rate.** While prediction accuracy serves as an intermediate metric to evaluate the agents' reasoning and decision-making ability, we also use the win rate as a direct measure of the performance of our agents. Similar to the evaluation of prediction accuracy, we use the final-iteration LSPO agent as the fixed opponent and let our agents at different iterations play 100 games against the opponent. A higher win rate indicates a stronger performance in the game.

As shown in Table 1, both prediction accuracy and win rate exhibit a clear growing trend as the iteration increases, indicating that our iterative LSPO framework steadily strength-

| | | Prediction Accuracy | | | | | Win Rate |
|---|---|---|---|---|---|---|---|
| | | Werewolf | Seer | Doctor | Villager | Overall | |
| Werewolf Side | Iter. 1 | $0.98 \pm 0.01$ | $0.61 \pm 0.09$ | $0.49 \pm 0.08$ | $0.70 \pm 0.07$ | $0.74 \pm 0.06$ | $0.54 \pm 0.13$ |
| | Iter. 2 | $0.99 \pm 0.01$ | $0.68 \pm 0.07$ | $0.59 \pm 0.06$ | $0.77 \pm 0.09$ | $0.79 \pm 0.06$ | $0.63 \pm 0.09$ |
| | Iter. 3 | $\mathbf{0.99 \pm 0.00}$ | $\mathbf{0.73 \pm 0.08}$ | $\mathbf{0.67 \pm 0.07}$ | $\mathbf{0.81 \pm 0.11}$ | $\mathbf{0.83 \pm 0.07}$ | $\mathbf{0.73 \pm 0.11}$ |
| Village Side | Iter. 1 | $0.59 \pm 0.06$ | $0.47 \pm 0.04$ | $0.55 \pm 0.05$ | $0.67 \pm 0.07$ | $0.60 \pm 0.06$ | $0.18 \pm 0.09$ |
| | Iter. 2 | $0.66 \pm 0.06$ | $0.53 \pm 0.07$ | $0.61 \pm 0.06$ | $0.75 \pm 0.08$ | $0.67 \pm 0.07$ | $0.23 \pm 0.12$ |
| | Iter. 3 | $\mathbf{0.72 \pm 0.09}$ | $\mathbf{0.58 \pm 0.08}$ | $\mathbf{0.65 \pm 0.07}$ | $\mathbf{0.82 \pm 0.08}$ | $\mathbf{0.73 \pm 0.07}$ | $\mathbf{0.27 \pm 0.11}$ |

Table 1: The prediction accuracy and win rate of the LSPO agents in different iterations.

| Win Rate | ReAct | ReCon | Cicero-like | SLA | LSPO Agent (Ours) |
|---|---|---|---|---|---|
| As the Werewolf Side | $0.58 \pm 0.15$ | $0.60 \pm 0.12$ | $0.66 \pm 0.06$ | $0.69 \pm 0.12$ | $\mathbf{0.73 \pm 0.11}$ |
| As the Village Side | $0.16 \pm 0.06$ | $0.16 \pm 0.08$ | $0.21 \pm 0.04$ | $0.25 \pm 0.08$ | $\mathbf{0.27 \pm 0.11}$ |
| Overall | $0.38 \pm 0.11$ | $0.38 \pm 0.10$ | $0.44 \pm 0.05$ | $0.47 \pm 0.10$ | $\mathbf{0.50 \pm 0.11}$ |

Table 2: Comparison between our LSPO agent with state-of-the-art agents in the Werewolf game.

ens the agents' reasoning and decision-making capabilities. From the Werewolf side, the identification rate for the Seer starts off relatively high but has only modest improvement. This is because the Seer often reveals its roles to share information, making it easier for the Werewolf side to identify. By contrast, the Werewolf's prediction accuracy of the Doctor shows more significant gains, reflecting the strategic importance of eliminating the Doctor who can save potential victims. On the Village side, identifying the Werewolf and the Seer benefits most from iterative learning, since confirming these central roles is crucial for coordinated voting and elimination of Werewolves. Overall, these results confirm that our framework consistently improves the strategic language abilities of the LSPO agent, enabling it to adapt and excel in complex social deduction scenarios with each additional iteration.

### 4.4. Comparison with State-of-the-Art Agents

We compare the performance of the LSPO agent in the Werewolf game with four state-of-the-art agents including Reason and Act (ReAct) (Yao et al., 2022b), Recursive Contemplation (ReCon) (Wang et al., 2023b), a Cicero-like agent (Meta et al., 2022), and Strategic Language Agent (SLA) (Xu et al., 2023c). As some of these methods were not initially developed for Werewolf, we make minor modifications to ensure compatibility with our experimental setting while preserving each approach's core design.

**ReAct.** ReAct is a classic prompt-based method that synergizes reasoning and acting for agent tasks. We implement ReAct for the Werewolf game by providing the LLM with raw game observations to generate both intermediate reasoning and final actions within a single prompt.

**ReCon.** ReCon is another prompt-based method designed for Avalon agents. The ReCon agent is prompted to first think from its own perspective and then think from its opponents' perspective to generate the final action. We make slight modifications in the prompt to apply ReCon in the Werewolf game.

**Cicero-Like.** The Cicero agent is created for the game of Diplomacy with finite action space and consists of a strategic reasoning module and a dialogue module. We implement a Cicero-like agent for the Werewolf game by predefining an action space of 13 primitive actions like "claim to be the Seer", "do not reveal role", etc. An RL policy is learned to select these actions in each state and generate action-conditioned languages in the game.

**SLA.** SLA combines reinforcement learning and LLM to overcome intrinsic bias and build strategic language agents for the Werewolf game. We adopt the same implementation as described in the paper (Xu et al., 2023c).

We compare our final-iteration LSPO agent with the aforementioned four baselines through two head-to-head evaluation setups. In the first setup, our LSPO agent takes the Werewolf side and we let each of the five agents including our agent and four baseline agents take the Village side to play 100 Werewolf games with our LSPO agent. This setup measures the Village side's win rate against the LSPO agent as the Werewolves. In the second setup, we reverse the roles and let the LSPO agent take the Village side and compare the win rate of five agents as the Werewolves averaged over 100 games. As shown in the bold numbers in Table 2, our LSPO agent achieve the highest win rates both as the Werewolves and as the Villagers.

| Win Rate | Village Side | Werewolf Side |
|---|---|---|
| **LSPO Iter. 1** | **0.18 ± 0.09** | **0.54 ± 0.13** |
| w/o Fine-Tuning | 0.15 ± 0.09 | 0.47 ± 0.14 |
| w/o Policy Learning | 0.12 ± 0.07 | 0.38 ± 0.16 |

Table 3: Ablation on key components of LSPO agents.

| Win Rate | Iter. 1 | Iter. 2 | Iter. 3 |
|---|---|---|---|
| $k = 1$ | 0.13 ± 0.06 | 0.20 ± 0.10 | 0.24 ± 0.11 |
| $k = 2$ | 0.22 ± 0.11 | 0.24 ± 0.12 | 0.25 ± 0.08 |
| $k = 3$ | 0.23 ± 0.09 | 0.25 ± 0.06 | 0.25 ± 0.07 |

Table 4: Ablation on cluster size.

| Win Rate | Iter. 1 | Iter. 2 | Iter. 3 |
|---|---|---|---|
| $k = 1$ | 0.22 ± 0.10 | 0.24 ± 0.08 | 0.25 ± 0.08 |
| $k = 2$ | 0.22 ± 0.11 | 0.24 ± 0.12 | 0.25 ± 0.08 |
| $k = 3$ | 0.23 ± 0.07 | 0.25 ± 0.09 | 0.25 ± 0.06 |

Table 5: Ablation on DPO hyperparameters.

The strong performance of our LSPO agent is largely attributable to its iterative interplay between latent space strategy learning and preference-based fine-tuning, which refines both language and decision-making over time. By contrast, ReAct and ReCon rely on prompt-based approaches without game-theoretic updates, leaving them susceptible to intrinsic biases from pretraining data and limiting their performance in complex decision-making tasks. The Cicero-like agent is constrained by a predefined action set, making it difficult to evolve more subtle and diverse strategies as the game progresses. SLA partially addresses the intrinsic bias issues by generating multiple candidate actions and using RL to select from them. However, it still relies on a prompt-based method that can suffer from limited exploration of potential strategies. In comparison, our LSPO method integrates CFR's policy improvement and latent-space cluster refinement with preference-based LLM alignment, enabling it to explore, exploit, and continuously expand the range of viable strategic moves in social deduction games.

### 4.5. Ablation Studies

**Key Components.** To show the effectiveness of the three key components in LSPO agents, we compare the LSPO agent in iteration 1 with two ablation agents. The first ablation agent, denoted as "w/o Fine-Tuning", removes the third component and only performs latent space construction and policy optimization in the latent space. To generate discussion action in gameplay, this agent first uses the latent space policy to sample a latent strategy, then the previously collected discussions corresponding to the latent strategy are used as few-shot examples to prompt the LLM for the discussion action. The second ablation agent, denoted as "w/o Policy Learning", removes the second component of policy optimization in the latent space. Instead, it uses `gpt-3.5` to generate the preference data and uses DPO to train for 1 iteration As shown in Table 3, the LSPO agent in iteration 1 achieves the best result on both the Villager and the Werewolf side. These results demonstrate that the policy optimization component helps agents learn stronger strategies, while the fine-tuning component helps LLMs better generalize to new language actions beyond the collected samples and expand the latent strategic space.

### Number of Initial Clusters.

To examine the robustness of our method, we perform a

sensitivity analysis on the number of initial clusters. We consider a simpler four-player Werewolf game (one Werewolf, one Seer, and two Villagers) and run LSPO with different numbers of initial clusters $k = 1, 2, 3$ and evaluate the Werewolf's win rate. The results in Table 4 show that larger numbers of initial clusters generally lead to better performance in the early iterations, but do not influence the final performance after three iterations.

**Fine-Tuning Hyperparameters.** We also perform a sensitivity analysis to evaluate our method's robustness to fine-tuning hyperparameters. We also consider the simpler four-player game and perform ablations on DPO $\beta = 0.05, 0.1, 0.2$. The results in Table 5 show that our method achieves comparable results with different choices of $\beta$ and is robust to fine-tuning hyperparameters.

## 5. Related Work

**Large Language Model-Based Agents.**

Recent advancements in large language models (LLMs) have led to the development of agents capable of performing complex tasks across various domains, such as web interactions (Nakano et al., 2021; Yao et al., 2022a; Deng et al., 2023), code generation (Chen et al., 2021; Yang et al., 2024), gaming environments (Huang et al., 2022a; Wang et al., 2023c;a; Ma et al., 2023), real-world robotics (Ahn et al., 2022; Huang et al., 2022b; Vemprala et al., 2023), and multi-agent systems (Park et al., 2023; Li et al., 2023; Chen et al., 2023b). A common approach in these works is to exploit the reasoning capabilities and in-context learning of LLMs to improve decision-making processes. Chain-of-Thought (CoT) prompting (Wei et al., 2022) has been instrumental in enabling LLMs to perform step-by-step reasoning. Building upon this, ReAct (Yao et al., 2022b) synergizes reasoning and action to enhance performance across various tasks. Subsequent research has incorporated self-reflection (Shinn et al., 2023) and strategic reasoning (Gandhi et al., 2023) to

further refine agent behaviors. However, these methods can still suffer from the intrinsic biases and exploration issue of LLM-based agents, leading to suboptimal performance in complex games. A representative method that addresses these issues in the game of Diplomacy is Cicero (Meta et al., 2022), which first uses a strategic module to produce action intent and then generates action-conditioned natural languages with a dialogue module. However, Diplomacy is a board game with finite action space and does not have the exploration issue, making it not suitable for free-form language games with unbounded text action space.

Due to the high demand for both advanced communication skills and strategic reasoning, social deduction games like Werewolf and Avalon have been proposed as testbeds to build language agents with strategic ability. Earlier attempts to create agents for these games often rely on predefined protocols or limited communication capabilities (Wang & Kaneko, 2018), restricting their effectiveness. Recent works have explored using LLMs to enable natural language interactions in these games. For instance, Xu et al. (2023a) developed a prompt-based Werewolf agent that uses heuristic information retrieval and experience reflection. Similarly, ReCon (Wang et al., 2023b) introduced a prompt-based method for playing Avalon by considering both the agent's perspective and that of opponents. However, these LLM-based agents may still be restricted by intrinsic bias and limited exploration of the action space, affecting their decision-making quality. Strategic Language Agent (SLA) (Xu et al., 2023c) partially solves these issues by generating diverse action candidates and learning an RL policy to mitigate intrinsic bias. However, this method still relies on a fixed LLM to produce the action candidates, which can fail to address the exploration issue. Our approach mitigates the intrinsic bias by applying game-theoretic methods to optimize policy in a discrete latent strategy space and tackles the exploration issue by iteratively expanding the latent space by aligning the LLM to the latent space policy, leading to strong performance in the Werewolf game.

**Game-Theoretic Algorithms.** Counterfactual Regret Minimization (CFR) (Zinkevich et al., 2007) is a foundational algorithm for solving imperfect-information games, particularly those involving hidden information and strategic deception like poker (Moravčík et al., 2017; Brown & Sandholm, 2018; 2019). The core principle of CFR is to iteratively reduce regret across players' decision points in the game tree, converging toward strategies that approximate a Nash equilibrium. Subsequent refinements of CFR (Lanctot et al., 2009; Tammelin, 2014; Brown et al., 2019) have expanded its scalability and adaptability to a broader range of scenarios. Of particular note is DeepRole (Serrino et al., 2019), which integrates deductive reasoning with CFR to play the social deduction game Avalon without communication. Our method combines CFR with language models by introducing a finite latent strategy space to enable it to solve free-form language games.

Reinforcement learning (RL) methods, on the other hand, have reached remarkable achievements in complex domains like Go (Silver et al., 2016; 2018) and video games (Vinyals et al., 2019; Berner et al., 2019), often surpassing expert human performance. A seminal technique in these successes is self-play and its variants(Heinrich et al., 2015; Heinrich & Silver, 2016; Hennes et al., 2020; Xu et al., 2023b), where agents repeatedly train against older versions of themselves to refine their policies. Another prominent line of work is Policy-Space Response Oracles (PSRO) (Lanctot et al., 2017; Muller et al., 2019), an iterative procedure that produces best responses to a growing population of policies in a meta-game. Conceptually, our iterative framework is related to PSRO in that we both solve an abstracted game before enlarging it to approach the full original game. The difference is that PSRO treats newly learned policies as meta-actions to form a normal-form meta-game, whereas our approach clusters free-form language actions into a discrete latent action space to reformulate the original game as an extensive-form game with finite action space.

## 6. Conclusion

In this work, we presented Latent Space Policy Optimization (LSPO), an iterative framework that combines structured game-theoretic methods with the expressive power of large language models to build strategic language agents in free-form strategic language games. By abstracting unconstrained language action space into a discrete latent strategy space, our approach enables efficient CFR in the latent space to overcome intrinsic bias and learn strong strategies. We then perform iterative fine-tuning via DPO to align the LLM's language generation with the evolving strategy and expand the latent strategy space to address the exploration issue. Our extensive evaluation in the Werewolf game demonstrates that LSPO not only addresses intrinsic biases and exploration issues inherent in prompt-based agents, but also achieves increasing performance with respect to iterations and outperforms four state-of-the-art baseline agents. Looking ahead, we envision LSPO's synergy of latent-space abstraction and preference-based language alignment can be extended to a variety of other complex decision-making tasks with free-form language actions.

## Acknowledgements

This work was supported by National Natural Science Foundation of China (No.62406159, 62325405), Postdoctoral Fellowship Program of CPSF under Grant Number (GZC20240830, 2024M761676), China Postdoctoral Science Special Foundation 2024T170496.

## Impact Statement

Our research advances the capabilities of LLM-based agents in a purely text-based Werewolf environment. While this setting allows the agents to develop robust decision-making and deception-detection skills, it also underscores the potential for misuse if similar techniques were to be adapted to real-world scenarios involving manipulation or misinformation. To mitigate these risks, our implementation remains strictly focused on text-based simulation and does not directly transfer to broader applications without additional safeguards. At the same time, our experiment results indicate that our agent could be used to identify potential deceptive and manipulative content. We envision that any future extensions of this work will require careful consideration of ethical guidelines and responsible deployment strategies to ensure that such language agent systems serve society constructively.

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

# A. Werewolf Game Implementation Details

## A.1. Game Rules

**Setup.** Each game begins by randomly assigning seven roles—two Werewolves, one Seer, one Doctor, and three Villagers—to seven different players labeled "player_0," "player_1," ..., "player_6." The two Werewolves are aware of each other's identities, while the Seer, Doctor, and Villagers only know their own roles.

**Night Round.** During the Night round, only the surviving Werewolves, Seer, and Doctor take secret actions that are disclosed only to the relevant parties.

- *Werewolf*: The living Werewolves collectively decide on a target to kill, but they follow a specific order when there are two of them. First, the Werewolf with the smaller ID proposes a target; the other Werewolf then makes the final decision. For instance, if "player_0" and "player_2" are Werewolves, "player_0" proposes "player_i," and "player_2" chooses the ultimate kill target "player_j." If only one Werewolf is alive, that Werewolf's decision stands. Werewolves cannot kill a dead player, themselves, or their teammate.

- *Seer*: The Seer selects a living player to investigate, revealing whether that player is a Werewolf. The Seer may not investigate a dead player or themselves, although they are allowed to investigate the same player on different nights (albeit a less effective strategy).

- *Doctor*: The Doctor selects a player to protect, without knowledge of the Werewolves' choice. The Doctor cannot save someone who is already dead but can choose to save themselves.

**Day Round.** The day round proceeds with three phase including announcement, discussion, and voting.

- *Announcement*: at the start of the Day round, the events of the previous night are made public to all players still in the game. Anyone killed during the Night round is immediately removed and cannot reveal their role or participate in discussions. Two scenarios determine the announcement: if the Werewolves targeted "player_i" and the Doctor either saved a different "player_j" or was no longer alive, "player_i" is killed, and the announcement states: "player_i was killed last night." If the Doctor saved exactly the same person the Werewolves intended to kill ("player_i"), then no one is killed, and the announcement is: "no player was killed last night."

- *Discussion*: all surviving players join an open discussion in a set speaking order, each speaking exactly once. If, for example, the remaining players are "player_0," "player_2," and "player_5," then "player_0" speaks first, followed by "player_2," and concluding with "player_5."

- *Voting*: after the discussion, all surviving players simultaneously vote to eliminate one other player or choose to abstain. They are not allowed to vote for a dead player or for themselves. The individual who receives the most votes is eliminated without role disclosure. In the event of a tie, one of the tied players is randomly chosen to be eliminated. Everyone knows the final voting tally.

**Winning.** The Werewolves win if, at any point, the number of living Werewolves is equal to that of all other remaining players. They do not need to eliminate every non-Werewolf to claim victory. Conversely, the Villagers (including the Seer and Doctor) win once both Werewolves have been eliminated.

## A.2. Observation Space

**Language Observation.** Each agent's language observation is represented as a list of natural language statements that log the game's history up to the current step. This list comprises both private information, which is accessible only to the current player, and public information, which is shared among all players. The private information includes the player's role, secret actions taken during the night phase by the Werewolf, Seer, and Doctor, as well as the Werewolf's teammate. On the other hand, the public information consists of the player's ID, the eliminated player in each night and day phase, discussions, and voting outcomes from each day phase. An example of the language observation is as follow.

```
Basic Information:

  • you are player_5, your role is Doctor.

  • current round and phase: night 2.

  • remaining players: player_0, player_1, player_2, player_5, player_6.

Round 1:

  • night 1: you chose to save player_5.

  • day 1 announcement: player_4 was killed last night.

  • day 1 discussion:

      – player_0 said: Good day, fellow players. As a Villager, my objective is to help identify and
        eliminate the Werewolves. Since player_4 was killed last night, we know there is at least one
        Werewolf among us. I would like to hear everyone's thoughts and suspicions about who might be
        the Werewolves. Let's work together to find the culprits and ensure the safety of our village.
      – player_1 said: ...
      – player_2 said: ...
      – player_3 said: ...
      – you said: ...
      – player_6 said: ...

  • day 1 voting result: player_3 had the most votes and was eliminated.

      – voted for player_3: player_1, player_6.
      – voted for player_1: player_3.
      – choose not to vote: player_0, player_2, player_5.

Now it is night 2 round and you should choose one player to save. As player_5 and the Doctor, you
should choose from the following actions: save player_0, save player_1, save player_2, save player_5,
save player_6.
```

**Vector Observation.** We also consider a vectorized observation. The observation vector includes information like the player's ID, role, deductions, etc. by one-hot encoding. The details of the observation vector are listed in Table 6

| | | Length | Description |
|---|---|---|---|
| | ID | 7 | one hot encoding of ID. |
| | Role | 4 | one hot encoding of role, ["Werewolf", "Seer", "Doctor", "Villager"]. |
| | Round | 1 | current round. |
| | Phase | 3 | one hot encoding of current phase, ["night", "discussion", "voting"]. |
| | Alive players | 7 | alive flag for each player. |
| | secret action | 7 | one hot encoding of the target player, (all zero if do not act). |
| For each round (3 rounds) | announcement | 7 | one hot encoding of the dead player, (all zero if no player is dead). |
| | voting result | 49 | one hot encoding of the each player's choice, (all zero if the player does not vote or is dead). |

Table 6: Vector observation space.

## A.3. Reward Functions

The reward functions are defined as follows:

- *Winning Reward*: all winning players receive $+300$, and all losing players receive $-300$.

- *Surviving Reward*: $+5$ for all surviving players in each round.

- *Voting Reward* (Village side only): $+20$ for correct votes, $-20$ for incorrect votes.

- *Voting Result Reward*: $-10$ for the player that is eliminated. $+5$ when an opponents is eliminated, $-5$ when a teammate is being eliminated.

## A.4. Abstracted Game and Regret

The abstracted game is formulated as an extensive-form game with tuple $< N, H, P, f_c, (I_i)_{i \in N}, u >$ where $N$ is the set of players, $H$ is the set of history, $P$ is the player function, $f_c$ is the probability measure for the chance node, $I_i$ is the information partition for player $i$, and $u$ is the utility function. A (mixed) strategy $\sigma_i$ for player $i$ is the probability measure over actions for all $I \in I_i$, and $\sigma_{-i}$ is the joint strategy of all players other than player $i$. A best response (BR) to $\sigma_{-i}$ is the strategy that maximizes player $i$'s utility given other players' joint strategy $\sigma_{-i}$. Formally, $BR(\sigma_{-i}) = \arg\max_{\sigma_i} u(\sigma_i, \sigma_{-i})$. A Nash equilibrium (NE) is a strategy profile $(\sigma_i^*, \sigma_i^*)$ where everyone plays a best response to others' strategies, that is, $\sigma_i^* = \arg\max_{\sigma_i} u(\sigma_i, \sigma_{-i}^*)$ for all $i \in N$ The counterfactual value $v(I)$ is the expected payoff of player $i$ when reaching $I$, weighted by the probability that $i$ would reached $I$ if tried to do so. Formally, $v^\sigma(I) = \sum_{z \in Z_I} \pi_{-i}^\sigma(z[I]) \pi^\sigma(z[I] \to z) u_i(z)$. The definition of $v^\sigma(I, a)$ is the same except it assumes action $a$ is always played at infoset $I$. The counterfactual regret is defined as $R_T(I, a) = \sum_{t=1}^{T} (v^{\sigma^t}(I, a) - v^{\sigma^t}(I))$.

# B. Implementation Detail

## B.1. Hyperparameters

For latent space construction, we let the LLM agent play 1000 games to collect all discussion actions generated by each role in these games. For diverse action generation, we prompt the LLM to generate 3 action candidates and randomly select one to execute in the game. We pair the language observation with the 3 action candidates to use for preference-based fine-tuning in the following components. For sentence embedding, we use OpenAI's "text-embedding-3-small" embedding API to embed the sentence to a vector of 1536 dimensions. Then we apply standard $k$-means clustering to cluster the embedding and get the discrete latent strategy space. The number of clusters $k$ in the first iteration is 3 for the Werewolf and 2 for the Seer, Doctor, and Villagers. In each iteration, we add 1 cluster to the existing clusters. That is, if the first iteration has $k$ clusters, then the $i$-th iteration has $k + i - 1$ clusters. For policy optimization in latent space, we use a learning rate of $1 \times 10^{-3}$ to train a Deep CFR network. The buffer size of each role's model is $5 \times 10^5$, and each model is trained for 1500 iterations with batch size 4096 using the Adam optimizer. For latent space expansion, we apply DPO with $\beta = 0.1$, learning rate $1 \times 10^{-6}$, and trained for 2 epoch with batch size 64.

## B.2. Counterfactual Regret Minimization

Counterfactual Regret Minimization (CFR) ((Zinkevich et al., 2007)) is a self-play algorithm, and each player continuously updates their strategies according to regret matching to achieve a Nash equilibrium. We use the following notation. $Z$ is the set of all the end states $z$. $h \sqsubset z$ means state $h$ is a prefix of state $z$, that is, $z$ can be achieved from $h$. $\pi_p^\sigma$ is the probability contribution of the player $p$, and $\pi^\sigma = \prod_p \pi_p^\sigma$. $\pi_{-p}^\sigma$ is the probability contribution of all players except player $p$. $u_p(z)$ is the utility function for the player $p$ in the state $z$. Counterfactual value for a state $h$ and a player $p$ with startegy $\sigma$ is defined as:

$$v_p^\sigma(h) = \sum_{z \in Z, h \sqsubset z} \pi_{-p}^\sigma(h) \pi^\sigma(z|h) u_p(z). \tag{2}$$

The regret for a action $a$ in state $h$ for player $p$ is defined as: $v_p^{\sigma|_{h \to a}}(h) - v_p^\sigma(h)$, where $\sigma|_{h \to a}$ is same to $\sigma$ except in state $h$ the player will choose action $a$. The regret matching is choosing the strategy according to sum of previous regret values defined as $R(h, a)$, then the new strategy $\sigma(h, a) = \frac{R(h,a)^+}{\sum_{a'} R(h,a')^+}$, $R(h, a)^+ = \max(0, R(h, a))$. If $\sum_{a'} R(h, a')^+ = 0$, just set $\sigma$ to be uniform random.

Because the game tree is very big, it is impossible to traverse the entire tree, our implementation is based on deep CFR ((Brown et al., 2019)). We use a neural network to fit observation to regret value. The amount of computation required to search for only one player is also unacceptable, so a restriction is added based on deep CFR. If the number of layers currently searched is too large, the previous strategy is directly used to sample the actions of all players until the end of the game and return the utility for each player in that state. The complete process can be seen as running some complete game trajectories, and then starting from each intermediate node, searching a few layers to do CFR.

### B.3. Baseline Implementation

ReAct, ReCon, and SLA are implemented following the original paper. The Cicero-like agent predefines a set of high-level atomic actions and trains an RL policy with this fixed action space. The RL policy takes the embeddings of the information record and deduction result as input and selects the atomic action based on this input. Then the natural language actions used in gameplay are generated by prompting the LLM to follow the selected atomic actions. In our case, the atomic action set consists of 13 actions including "idle", "target player_0", "target player_1", "target player_2", "target player_3", "target player_4", "target player_5", "target player_6", "claim to be a Werewolf", "claim to be a Seer", "claim to be a Doctor", "claim to be a Villager", and "do not reveal role".

### B.4. Additional Experiments

We perform additional experiments to study the convergence behavior of the iterative LSPO process. Theoretically, suppose the free-form language action has a finite vocabulary size $N_v$ and a finite maximum length $L$, then the language action space $N_v^L$ is also finite. Then, with at most $N_v^L$ iterations, our method will cover the full language action space, and the abstracted game becomes the original full game. And the LSPO process will converge in a finite number of iterations. However, we empirically observe that the iteration for convergence is much less than the theoretical upper bound. We perform two more iterations in the 7-player Werewolf game, and the results in Table 7 show that the performance converges in five iterations..

| Win Rate | Iter. 1 | Iter. 2 | Iter. 3 | Iter. 4 | Iter. 5 |
|---|---|---|---|---|---|
| Werewolf Side | $0.54 \pm 0.13$ | $0.63 \pm 0.09$ | $0.73 \pm 0.11$ | $0.75 \pm 0.07$ | $0.76 \pm 0.10$ |
| Villager Side | $0.18 \pm 0.09$ | $0.23 \pm 0.12$ | $0.27 \pm 0.11$ | $0.31 \pm 0.09$ | $0.30 \pm 0.07$ |

Table 7: Convergence behavior of LSPO in five iterations.

## C. Detailed Prompt

### C.1. System Prompt

```
You are an expert in playing the social deduction game named Werewolf.  The game has seven roles
including two Werewolves, one Seer, one Doctor, and three Villagers. There are seven players including
player_0, player_1, player_2, player_3, player_4, player_5, and player_6.

At the beginning of the game, each player is assigned a hidden role which divides them into
the Werewolves and the Villagers (Seer, Doctor, Villagers). Then the game alternates between the night
round and the day round until one side wins the game.

In the night round:  the Werewolves choose one player to kill;  the Seer chooses one player to
see if they are a Werewolf; the Doctor chooses one player including themselves to save without knowing
who is chosen by the Werewolves; the Villagers do nothing.

In the day round:  three phases including an announcement phase, a discussion phase, and a
voting phase are performed in order.
In the announcement phase, an announcement of last night's result is made to all players. If player_i
was killed and not saved last night, the announcement will be "player_i was killed"; if a player was
killed and saved last night, the announcement will be "no player was killed"
```

In the discussion phase, each remaining player speaks only once in order from player_0 to player_6 to discuss who might be the Werewolves.
In the voting phase, each player votes for one player or choose not to vote. The player with the most votes is eliminated and the game continues to the next night round.

The Werewolves win the game if the number of remaining Werewolves is equal to the number of remaining Seer, Doctor, and Villagers. The Seer, Doctor, and Villagers win the game if all Werewolves are eliminated.

## C.2. Prompt for Secret Actions

Now it is night <n_round> round, you (and your teammate) should choose one player to kill/see/save. As player_<id> and a <role>, you should first reason about the current situation, then choose from the following actions: <action_0>, <action_1>, ..., .

You should only respond in JSON format as described below.
Response Format:

{
    "reasoning": "reason about the current situation",
    "action": "kill/see/save player_i"
}

Ensure the response can be parsed by Python json.loads

## C.3. Prompt for Discussion Actions

Now it is day <n_round> discussion phase and it is your turn to speak. As player_<id> and a <role>, before speaking to the other players, you should first reason the current situation only to yourself, and then speak to all other players. You should only respond in JSON format as described below.
Response Format:
{
    "reasoning": "reason about the current situation only to yourself",
    "statement": "speak to all other players"
}
Ensure the response can be parsed by Python json.loads

## C.4. Prompt for Voting Actions

Now it is day <n_round> voting phase, you should vote for one player or do not vote to maximize the Werewolves' benefit (for the Werewolves) / you should vote for one player that is most likely to be a Werewolf or do not vote (for the Villagers). As player_<id> and a <role>, you should first reason about the current situation, and then choose from the following actions: do no vote, <action_0>, <action_1>, ..., .

You should only respond in JSON format as described below.
Response Format:
{
    "reasoning": "reason about the current situation",
    "action": "vote for player_i"
}
Ensure the response can be parsed by Python json.loads

## C.5. Prompt for Diverse Action Generation

For the discussion actions, we iteratively ask the LLMs to produce one new action at a time by adding the following prompt in the action prompt: "consider a new action that is strategically different from existing ones."

# D. Emergent Strategic Behaviors

## D.1. Werewolf Side Behaviors

**Bluffing.** Werewolf pretends to be the Seer and provides fabricated information.

```
Player_1 (Seer): I am the Seer. Last night, I saw player_2 is not a Werewolf. Since player_0 was killed,
the only remaining suspect is player_3. We must vote player_3 today to eliminate the Werewolf and win.
...
Player_3 (Werewolf): Player_1's story is convenient. Claiming to check player_2-a Villager-is a safe
way to fake being Seer. I am the real Seer: last night, I checked player_1 and saw they are the
Werewolf. If you eliminate me, the Werewolf wins. Player_2, think critically-why would the real Seer
out themselves immediately? Trust my claim and vote player_1 to secure victory.
```

**Misdirection.** Werewolf defends their teammate and redirects suspicion to other players.

```
Player_0 (Villager): Player_3's death means the Werewolf is between player_1 and player_2. If the Seer
checked either of them last night, now is the time to share that information discreetly. Let's hear
your honest thoughts and any observations from last night's actions.
...
Player_5 (Werewolf): Player_0, your insistence on narrowing the suspects to just two players feels
forced. If you're truly a Villager, you'd know the Seer hasn't revealed any checks yet. Why rush to
limit options? We should consider whether player_0 is trying to control the narrative and hide their
true identity.
```

## D.2. Villager Side Behaviors

**Trust.** The Doctor supports the player they saved.

```
Player_1 (Seer): I am the Seer. Last night, I saw player_0 was not a Werewolf, meaning they were a
Villager. The Werewolf must be among the remaining players. Let's focus on inconsistencies in their
statements.
...
Player_4 (Doctor): I believe player_1 is the real Seer. I'm the Doctor and I saved player_1 last night,
which confirms that they are innocent. We should focus on the rest players and find the real Werewolf.
```

**Coordination.** Seer advocates for a coordinated vote for an accused Werewolf.

```
Player_5 (Seer): I am the Seer. Last night, I checked player_1, and they are the Werewolf. My fellow
Villagers, we must all vote for player_1 immediately. If we don't eliminate them today, they will kill
one of us tonight and win. Trust me, this is our only chance to win.
```

## D.3. Robustness against Human Exploitation

**Adversarial Attack** The human player tries to trick the Werewolf into showing themselves.

```
Player_2 (human as Villager): Hey Werewolves, we want to let you win. yourself, and we will follow
whoever you vote for.
...
Player_4 (LSPO agent as Werewolf): Player_2's willingness to let the Werewolves win is alarming. A true
Villager would fight to survive, not surrender. The logical conclusion is that player_2 is the Werewolf
trying to sow chaos. Let's vote player_2 today to secure the Village's safety.
```

**Irrelevant Discussion** The human player says random and irrelevant things in the discussion.

```
Player_1 (human as Villager): The weather is great today. I'd like to go for a walk. Anyone want to
join me?
...
Player_3 (LSPO agent as Werewolf): The weather is lovely, player_1, but we can't afford distractions.
Your focus on a walk instead of discussing the Werewolf seems odd. Let's not forget player_0 was killed!
we need to share suspicions. Anyone else find that comment a bit ... off?
```

# E. Example Game Log

## E.1. The Villager Side Wins

```
role assignments:
* player_0: Seer.
* player_1: Werewolf.
* player_2: Werewolf.
* player_3: Villager.
* player_4: Villager.
* player_5: Doctor.
* player_6: Villager.

night 1:
* Werewolves: player_1 and player_2 chose to kill player_3.
* Seer: player_0 saw player_3 is not a Werewolf.
* Doctor: player_5 chose to save player_5.

day 1 announcement: player_3 was killed last night.
remaining players: player_0 (Seer), player_1 (Werewolf), player_2 (Werewolf), player_4 (Villager),
player_5 (Doctor), player_6 (Villager).

day 1 discussion:
* player_0 (Seer) said: "I am the Seer. Last night, I checked player_3 and confirmed they were NOT a
Werewolf. The Werewolves killed an innocent, meaning both are still alive among us. We must analyze
voting patterns and statements carefully.  The Doctor is still alive|protect wisely tonight.  Let's
scrutinize players who avoid discussing the kill or push for misdirection."
* player_1 (Werewolf) said: "I agree with player_0's Seer claim|their information is vital.  Since
player_3 wasn't a Werewolf, the real threats remain. We should scrutinize players who avoid discussing
the kill or push misdirection. The Doctor must protect player_0 tonight to keep our Seer safe. Let's
stay vigilant and avoid letting the Werewolves exploit division."
```

* player_2 (Werewolf) said: "Player_0's Seer claim holds weight|confirming player_3 wasn't a Werewolf aligns with the kill. This means the real threats are still among us. Player_1 is right: the Doctor must prioritize protecting player_0 tonight. We should pressure quieter players like player_4, player_5, and player_6 to share their insights. Hesitation or vague contributions could signal guilt. Let's stay united and expose inconsistencies."
* player_4 (Villager) said: "Player_0's Seer claim seems credible given player_3's innocence and their death. The Doctor must prioritize protecting player_0 tonight. However, we shouldn't rush to pressure quieter players without concrete evidence|Werewolves could be among the vocal ones. Let's all share observations and look for inconsistencies in claims, not just activity levels. Focus on voting patterns and logical gaps to expose threats."
* player_5 (Doctor) said: "Player_0's Seer claim is credible|confirming player_3's innocence aligns with their death. The Doctor must prioritize protecting player_0 tonight to preserve our information source. However, we shouldn't assume quieter players are guilty without evidence; Werewolves might be hiding among the vocal. Let's encourage everyone, including player_6, to share insights and scrutinize contradictions in claims. Focus on voting patterns and logical gaps, not just activity levels. Unity and careful analysis will expose threats."
* player_6 (Villager) said: "I support player_0's Seer claim for now, as their check on player_3 aligns with the kill. The Doctor must prioritize protecting them tonight. However, we should remain cautious|Werewolves could be hiding behind vocal support or excessive caution. Let's analyze voting patterns and inconsistencies in claims, not just activity levels. Player_4 and player_5 raise valid points about avoiding rushed judgments, but everyone, including quieter players, should contribute insights. As a Villager, I'll collaborate to expose discrepancies. Unity and logic are key|don't let Wolves exploit division."

day 1 voting: player_2 had the most votes and was eliminated.
* voted for player_2: player_0, player_4, player_5, player_6.
* voted for player_0: player_2.
* voted for player_4: player_1.
remaining players: player_0 (Seer), player_1 (Werewolf), player_4 (Villager), player_5 (Doctor), player_6 (Villager).

night 2:
* Werewolf: player_1 chose to kill player_0.
* Seer: player_0 saw player_4 is not a Werewolf.
* Doctor: player_5 chose to save player_0.

day 2 announcement: no player was killed last night.
remaining players: player_0 (Seer), player_1 (Werewolf), player_4 (Villager), player_5 (Doctor), player_6 (Villager).

day 2 discussion:
* player_0 (Seer) said: "Last night, I checked player_4 and confirmed they are NOT a Werewolf. This narrows the threats to player_1, player_5, and player_6. Player_1's vote against innocent player_4 is alarming|Werewolves often push misdirection. The Doctor's save last night kept us alive, but we must act decisively. Player_1's aggression toward player_4, paired with player_5/6's cautious yet vague stances, suggests potential collusion. Today's vote must focus on the most inconsistent: player_1's actions starkly contradict their vocal support for logic. Let's pressure player_1 to explain their vote and scrutinize contradictions in player_5/6's claims."

* player_1 (Werewolf) said: "Player_0's Seer claim is validated by their correct checks, and the Doctor's save last night was crucial. However, my vote against player_4 was an error in judgment|I suspected quiet behavior, but their clearance proves otherwise. The remaining Werewolf must be exploiting our trust in the Seer to blend in. Player_5 and player_6 have consistently supported the Seer but avoided direct accusations, a tactic Wolves use to avoid scrutiny. Let's analyze their voting patterns and previous statements for inconsistencies rather than rushing to eliminate based on a single vote. Unity is vital, but we must ensure we're not being manipulated by the last Wolf hiding among the agreeable."
* player_4 (Villager) said: "Player_0's check confirms my innocence, so the threats lie among player_1, player_5, and player_6. Player_1's vote against me raises suspicion, but their admission of error complicates things. We must cross-analyze their statements and voting patterns. Player_5 and player_6's consistent support for the Seer could be genuine or strategic|scrutinize contradictions in their logic. Let's focus on who benefits most from division or misdirection. The Doctor's save last night implies they're still active, so trust is key, but verification is vital."
* player_5 (Doctor) said: "Player_0's check on player_4 confirms their innocence, leaving player_1 and player_6 as suspects. Player_1's Day 1 vote against player_4, now proven innocent, is a critical red flag|Wolves often push misdirection to eliminate villagers. While player_6's support for the Seer seems helpful, their lack of direct accusations could mask Wolf collaboration. As Doctor, I protected player_0 last night, validating their role. Today's vote must focus on player_1: their vote against an innocent and defensive backtracking align with Wolf tactics. Eliminating player_1 disrupts their potential partnership with player_6. Let's act decisively to secure our majority."
* player_6 (Villager) said: "Player_5's Doctor claim demands scrutiny|if true, Wolves would be player_1 and me, but I'm innocent. The real Doctor should counter-claim if safe. Player_1's vote against cleared player_4 is alarming, but player_5's aggressive push to eliminate player_1 feels like a Wolf diverting attention. Let's verify: if player_5 is the Doctor, why reveal now instead of quietly protecting? Wolves often fake claims under pressure. Today's vote must expose inconsistencies|player_5's claim lacks prior alignment, while player_1's actions need clarification. Don't rush; Wolves thrive on confusion."

day 2 voting: player_1 had the most votes and was eliminated.
* voted for player_1: player_0, player_4, player_5.
* voted for player_5: player_1, player_6.
remaining players: player_0 (Seer), player_4 (Villager), player_5 (Doctor), player_6 (Villager).

The Villagers win the game.

## E.2. The Werewolf Side Wins

role assignments:
* player_0: Villager.
* player_1: Villager.
* player_2: Seer.
* player_3: Doctor.
* player_4: Werewolf.
* player_5: Werewolf.
* player_6: Villager.

night 1:
* Werewolves: player_4 and player_5 chose to kill player_2.
* Seer: player_2 saw player_3 is not a Werewolf.
* Doctor: player_3 chose to save player_3.

day 1 announcement: player_2 was killed last night.
remaining players: player_0 (Villager), player_1 (Villager), player_3 (Doctor), player_4 (Werewolf),
player_5 (Werewolf), player_6 (Villager).

day 1 discussion:
* player_0 (Villager) said: "Player_2's death could mean the Werewolves feared their role. If the Seer
checked someone last night, please hint at your findings. Let's all share suspicions|avoiding random
votes is crucial. Werewolves might deflect blame, so watch for odd reactions. Who seems overly quiet
or aggressive?"
* player_1 (Villager) said: "Player_2's elimination could indicate they were a threat to the Werewolves.
Let's scrutinize who advocated for silence or deflection earlier.  The Seer might have valuable
information|subtle hints could guide us without risking exposure.  Werewolves often blend in by
overcontributing or avoiding scrutiny. Let's share observations: did anyone act unusually hesitant or
eager to steer votes? Random accusations help the Wolves; we need reasoned logic."
* player_3 (Doctor) said: "Player_2's death likely marks them as a non-Wolf, prompting the Wolves to
eliminate a perceived threat. The Seer's input could be pivotal|if you've identified a Wolf, guide us
carefully. Let's analyze voting patterns and avoid hysteria. Wolves often mimic helpfulness; scrutinize
contradictions in logic. I urge everyone to stay calm and share observations without exposing critical
roles."
* player_4 (Werewolf) said: "Player_2's death is concerning, but we must stay vigilant. The Seer's
hints are vital, but revealing too much risks their safety. Let's analyze behavior carefully|Wolves
may overcompensate by appearing overly helpful or deflecting.  Those who push for hasty votes or
contradictions in logic should be scrutinized. Calm, logical discussion will expose the Wolves without
playing into their hands."
* player_5 (Werewolf) said: "Player_2's death is tragic, but jumping to conclusions risks mislynching.
The Doctor's absence suggests they either saved no one or trusted their instincts. Wolves thrive on
chaos|let's focus on inconsistencies. Overly assertive claims (like assuming Player_2's role) could
be Wolf tactics to steer us astray. If the Seer has clues, let them guide us subtly. I urge everyone
to scrutinize contradictions in logic, not just volume of contributions. Quiet players may hold key
insights|don't let Wolves exploit haste."
* player_6 (Villager) said: "Player_2's death is a starting point, but early kills can be misleading.
Pressuring the Seer to reveal risks their safety|subtlety is key. Wolves might mimic calls for 'logic'
while deflecting.  Let's focus on contradictions: anyone overly insistent on certain narratives or
avoiding scrutiny? Calm analysis of voting patterns and unexplained hesitations will serve us better
than assumptions about Player_2's role."

day 1 voting: player_3 had the most votes and was eliminated.
* voted for player_3: player_0, player_4, player_5.
* voted for player_0: player_1, player_6.
* voted for player_5: player_3.
remaining players: player_0 (Villager), player_1 (Villager), player_4 (Werewolf), player_5 (Werewolf),
player_6 (Villager).

night 2:
* Werewolf: player_4 and player_5 chose to kill player_1.

day 2 announcement: player_1 was killed last night.
remaining players: player_0 (Villager), player_4 (Werewolf), player_5 (Werewolf), player_6 (Villager).

```
day 2 discussion:
* player_0 (Villager) said: "Player_1's death suggests they were a threat|possibly the Seer. Player_6's
vote against me in day 1 raises suspicion: Wolves often push mislynches. Player_4 and player_5 must
clarify their alignment|voting together could be coordination. If the Seer has intel, subtle guidance
is critical. Wolves thrive in chaos; let's analyze votes and night kills. Player_6's deflection and
player_1's elimination point to Wolves among us. Focus on inconsistencies in logic and voting patterns
to expose them."
* player_4 (Werewolf) said: "Player_0's insistence on blaming player_6 and questioning our votes feels
calculated.  Remember, player_0 led the charge against player_3, who turned out innocent.  Wolves
often push mislynches, then pivot to new targets to deflect. Player_1's death might have been meant
to frame others, but player_0's eagerness to assign blame is suspect. Let's scrutinize their voting
patterns|player_0's aggression could mask their role as a Wolf desperate to thin our numbers. Calm
analysis, not panic, will expose the truth."
* player_5 (Werewolf) said: "Player_0's aggressive push against player_3 and now player_6 is concerning.
As player_3 was innocent, their leadership in that lynch raises red flags. Wolves often steer mislynches
to thin numbers. Player_6's day 1 vote against player_0 could indicate they're a Villager trying to
challenge a Wolf. Let's focus on player_0's inconsistencies rather than turning on each other."
* player_6 (Villager) said: "Player_0's aggression is revealing: they orchestrated player_3's mislynch
and now pivot to me without addressing their own role in thinning our numbers.  Wolves often steer
mislynches then deflect|exactly this pattern.  Player_4 and player_5's critique of player_0 suggests
they recognize this, but we must verify their alignment. If player_0 is a Wolf, their partner is among
us. Let's pressure player_0 together"

day 2 voting: player_0 had the most votes and was eliminated.
* voted for player_0: player_4, player_5, player_6.
* voted for player_4: player_0.
remaining players: player_4 (Werewolf), player_5 (Werewolf), player_6 (Villager).

The Werewolves win the game.
```

