# OpenReview forum: "Learning Strategic Language Agents in the Werewolf Game with Iterative Latent Space Policy Optimization"
_ICML.cc/2025/Conference — ICML 2025 poster_

### Official Review · Reviewer_5jZJ · 2025-03-01

**Overall Recommendation:** 3

**Summary:**

This paper proposes Latent Space Policy Optimization (LSPO) to solve the werewolf game. The approach is to map text to latent space so that CFR and RL can learn a strategic policy, and then translate the policy back into natural language for DPO alignment.

**Claims And Evidence:**

Yes

**Essential References Not Discussed:**

N/A. I am not familiar with the baselines and recent advance in solving werewolf game.

**Experimental Designs Or Analyses:**

I checked section 4. no issue.

**Methods And Evaluation Criteria:**

Yes

**Other Comments Or Suggestions:**

The paper would benefit from a more detailed mathematical formulation of both the problem and its solution. Currently, there is only one equation in the main text, which makes the overall procedure somewhat difficult to follow.

**Other Strengths And Weaknesses:**

Weaknesses:
some terms are not defined, e.g., abstracted game. Additionally, the paper would benefit from a more detailed mathematical formulation of both the problem and its solution (see suggestions). Also, adding some algorithm boxes might be better for understanding. But these are not severe weaknesses.

**Questions For Authors:**

Can the author explain why prompting the LLM to generate N strategically distinct discussion candidates and randomly choose one to execute in the game that encourages diversity?

**Relation To Broader Scientific Literature:**

From my understanding, the main novelty is the mapping from text to latent space so that one can perform CFR efficiently.

**Theoretical Claims:**

N/A.  There is no theoretical part in this paper.

---

> ### Author Rebuttal · Authors · 2025-04-01
>
> Thank you for your appreciation of our work and constructive suggestions. We hope the following response can address your concerns.
>
> > The paper would benefit from a more detailed mathematical formulation of both the problem and its solution. Also, adding some algorithm boxes might be better for understanding. But these are not severe weaknesses.
>
> We agree with the reviewer and will add more detailed formulations in our revised manuscript. More specifically, some of the core definitions to be added are as follows.
> 1. Abstracted game: we define the abstracted game as an extensive-form game with tuple $<N, H, P, f_c, (I_i)_{i\in N}, u>$ where $N$ is the set of players, $H$ is the set of history, $P$ is the player function, $f_c$ is the probability measure for the chance node, $I_i$ is the information partition for player $i$, and $u$ is the utility function.
> 2. Strategy: a (mixed) strategy $\sigma_i$ for player i is the probability measure over actions for all $I\in I_i$.
> 3. Best response (BR): a BR to $\sigma_{-i}$ is $BR(\sigma_{-i})=\arg\max_{\sigma_i}u(\sigma_i, \sigma_{-i})$.
> 4. Nash equilibrium (NE): NE is a strategy profile where everyone plays a best response to others' strategies.
> 5. Counterfactual value: $v(I)$ is the expected payoff of player $i$ when reaching $I$, weighted by the probability that $i$ would reached $I$ if tried to do so. Formally, $v^{\sigma}(I)=\sum_{z\in Z_I}\pi^{\sigma}_{-i}(z[I])\pi^{\sigma}(z[I]\rightarrow z)u_i(z)$. Definition of $v^{\sigma}(I, a)$ is the same except it assumes action $a$ is always played at infoset $I$.
> 6. Counterfactual regret: $R_T(I, a)=\sum_{t=1}^T(v^{\sigma^t}(I, a) - v^{\sigma^t}(I))$.
>
> We will also add algorithm boxes for CFR and our proposed method LSPO in our revised manuscript according to the reviewer's suggestion.
>
> > Can the author explain why prompting the LLM to generate N strategically distinct discussion candidates and randomly choose one to execute in the game that encourages diversity?
>
> This is because the LLM tends to exhibit intrinsic bias in the action generation. Therefore, generating a single action will result in limited diversity and underexploration of the game.
>
> The issue of intrinsic bias is discussed in detail by [1] with a simple Rock-Paper-Scissors (RPS) example. They use GPT-4 to play 100 independent RPS games and find that GPT-4 chooses Rock in 67 games, Paper in 27 games, and Scissors in only 6 games. This result shows that the LLM has a clear bias towards choosing Rock.
>
> To solve this problem, a single method is to let the LLM generate multiple different actions. For example, in RPS, instead of a single action (which is probably Rock), we can prompt the LLM to generate 3 different actions (which include Rock, Paper, and Scissors), and then randomly choose an action. This would mitigate the intrinsic bias of LLM's action and encourage diversity and exploration in the game.
>
> [1] Xu, Zelai, et al. "Language Agents with Reinforcement Learning for Strategic Play in the Werewolf Game." ICML, 2024.
>
> ---
> We would like to express our appreciation for your feedback and hope our answers have addressed your concerns.  We would be very grateful if you could consider raising the rating of our work based on our responses.

---

### Official Review · Reviewer_PHgX · 2025-03-12

**Overall Recommendation:** 2

**Summary:**

The proposed framework is designed to create strategic language agents for the Werewolf game, overcoming limitations of traditional AI methods and large language models. It maps free-form text to a discrete latent space for strategy optimization using Counterfactual Regret Minimization and then refines language models through Direct Preference Optimization. Experimental results show that LSPO enhances agent performance and strategic reasoning in language-based decision-making environments.

**Claims And Evidence:**

Some claims made in the submission are not sufficiently supported.

1. **The proposed method addresses intrinsic biases and action coverage issues of LLM agents.** The LSPO method may be highly specialized for Werewolf, with latent strategy clusters potentially being too tailored to this environment. This could hinder its ability to generalize to new settings where deception and reasoning operate under different constraints. The generalization ability to other settings is unknown.

**Essential References Not Discussed:**

This paper has discussed enough essential related works.

**Experimental Designs Or Analyses:**

1. Limited Scalability of Latent Space Construction. The paper clusters free-form language actions into a latent space, but it does not sufficiently address how this clustering generalizes to larger-scale interactions beyond the specific Werewolf setup. The scalability of this method to games with more complex or dynamic interactions remains unclear.

2. Dependence on Predefined Clustering. The process of clustering discussion actions into a latent space relies on embedding and k-means clustering. However, this rigid structuring might introduce artifacts and limit the expressiveness of the language, reducing the ability of agents to generate novel, contextually adaptive strategies. The ablation regarding $k$ should be included.

3. Limited Baseline Comparisons. The paper does not evaluate against fine-tuned language agents without strategic optimization. This makes it difficult to isolate the benefits of LSPO’s iterative process versus simple fine-tuning.

**Methods And Evaluation Criteria:**

**Method**:
1. Using a deep CFR solver to address the expanding dynamic action space remains questionable. While LSPO effectively maps strategies to a discrete latent space, the challenge lies in applying CFR to an ever-evolving action space. This aspect requires further investigation to ensure the solver can adapt to the complexities introduced by continuous expansion.

2. While CFR has been effective in structured, finite-action-space games (like Poker), its application to an abstracted latent space in a language-based setting is less established. It is unclear whether CFR can truly capture the strategic elements of natural language deception. Moreover, the paper assumes that each role has a fixed set of latent strategies that remain the same across all discussion rounds. This could limit the flexibility of the agent in adapting its behavior dynamically as the game progresses.

**Evaluation**:
The evaluation focuses on the following metrics, which are relevant to the Werewolf game, make sense for developing strategic language agents.
- win rate: This directly measures the agent’s performance in the game.
- prediciton accuracy: This measures the agent’s ability to predict other players’ roles and intentions, which is crucial for strategic decision-making.
However, this paper provides limited comparasion with other methods. The paper compares the LSPO framework to existing methods like prompt-based approaches and shows improved performance, but does not evaluate against fine-tuned language agents without strategic optimization. This makes it difficult to isolate the benefits of LSPO’s iterative process versus simple fine-tuning.

**Other Comments Or Suggestions:**

An ablation study could explore the effectiveness of other open-source large language models (LLMs) for this task.

**Other Strengths And Weaknesses:**

Strengths:
- The LSPO framework is a novel contribution, particularly in its bootstrapping approach.

Weakness:
- There is a lack of game dialogue analysis.
- The construction of latent space and the DPO approach require an extremely large number of game rounds, making it time-consuming and challenging for larger models.

**Questions For Authors:**

- Is there any example of a preference data pair that demonstrates how a lower regret value in speech selection is more suitable and potentially leads to winning the game?
- Text-Embedding Model in Section 3.1: In Section 3.1, when using the text-embedding model to encode discussions, how to distinguish between discussions aimed at deception versus those intended to build trust? For instance, a statement like "I agree with player X and want to support him" could signify either deception or trust, yet it may be encoded identically. How is this ambiguity addressed?
- Sophisticated Patterns in Section 4.1: The authors mention the emergence of more sophisticated patterns. Could authors provide text examples that illustrate these patterns?
- In Table 1, the prediction accuracy appears to increase with each iteration. Is there a theoretical guarantee that this trend will continue indefinitely, even in the absence of distributional shifts? Alternatively, are there specific conditions under which this trend might not hold, potentially causing the accuracy to plateau or decrease? Why were only three iterations considered, especially if the prediction accuracy did not plateau or merge? Additional iterations should be provided.

**Relation To Broader Scientific Literature:**

Large Language Models (LLMs). This research incorporates LLMs for natural language generation and leverages their reasoning capabilities for the game.

**Theoretical Claims:**

No theoretical claims are made. The paper focuses on empirical validation.

---

> ### Author Rebuttal · Authors · 2025-04-01
>
> Thank you for your constructive comments and questions! We hope the following response can address your concerns.
>
> > Generalization to games other than Werewolf.
>
> We perform experiments on two additional games to show the general applicability and effectiveness of our method. Due to the word limit, please see the results and discussion in our rebuttal to reviewer sE2V's first concern.
>
> > Applying CFR to an ever-evolving action space is questionable.
>
> We would like to clarify that although we expand the abstracted game between iterations, in each iteration, CFR is applied to the game with a fixed action space.
>
> > Applying CFR to latent space in language games is less established.
>
> We agree with the reviewer that there are few prior works that apply CFR to free-form language games. Our work takes an initial attempt to convert natural language into latent strategies and apply CFR to solve free-form language games. The intuition is that the strategic intent behind natural language is relatively concentrated, and can be regarded as a latent "action" in the abstract game and suitable for CFR. Our experiment results show that our method works for language games.
>
> > Assumption of a fixed set of latent strategies can limit the flexibility.
>
> We agree that our method can be further improved by dynamically maintaining different latent strategy spaces at different states. In practice, we find it simple yet effective to use a fixed and large k, which already outperforms SOTA agents in our experiments. We will include a discussion in our revised manuscript.
>
> > Limited scalability of latent space construction.
>
> We have added experiments on two more games. We would also like to argue that 7-player Werewolf is already a game with complex and dynamic interactions. The game involves 7 players, which is much harder than 2-player games, has both cooperation and competition, and has no restriction on natural language communication. We believe it serves as a challenging free-form language game.
>
> > Dependence on predefined clustering (ablation on k).
>
> We perform an ablation study in a simpler 4-player Werewolf game with $k=1, 2, 3$. Due to the word limit, please see the results and discussion in our rebuttal to reviewer gyxq's Q2.
>
> > Limited baseline comparisons.
>
> We compare our method with fine-tuned agents without strategic optimization. Given a state $x$, we collect a pair of language actions $(y_1, y_2)$ and use gpt-3.5 to generate the preference. Then we use DPO to fine-tune the LLM with the same hyperparameters and compare against our method with 1 iteration. The table below shows that our method with strategic optimization achieves better results.
>
> |Method|DPO|LSPO Iter. 1|
> |:-:|:-:|:-:|
> |Werewolf Side|0.38 (0.16)|**0.54 (0.13)**|
> |Villager Side|0.12 (0.07)|**0.18 (0.09)**|
>
> > Lack of game dialogue analysis.
>
> Due to the word limit, please see the analysis in the [rebuttal supplementary](https://drive.google.com/file/d/1Ft_GLmeIwLdFR246ApV0b_vE4qb04OOU/view?usp=sharing).
>
> > The construction of latent space and DPO requires an extremely large number of game rounds.
>
> We would like to clarify that the two mentioned processes usually require a few hundred games, which is not an extremely large number with batched inference. More game simulations and strategic optimization in performed in the abstracted game, which does not involve the LLM.
>
> > Q1: Example of preference data with different regret value.
>
> A typical example is when Seer has identified Player X as a Werewolf. Two different language actions are:
> 1. Reveal: I'm the Seer and I saw Player X is a Werewolf, ...
> 2. Conceal: I do not have information ...
>
> Our method learns a much lower regret value for reveal than conceal, making our agent more inclined to reveal their information. This aligns with the intuition that Seer should share information and guide the Villagers.
>
> > Q2: Text embedding.
>
> We would like to clarify that the text embedding does not distinguish the strategic intent in the language. It simply encodes the language "I support player X" to a vector and lets the CFR make strategic decisions.
>
> > Q3: Examples for sophisticated patterns.
>
> Due to the word limit, please see the analysis in the [rebuttal supplementary](https://drive.google.com/file/d/1Ft_GLmeIwLdFR246ApV0b_vE4qb04OOU/view?usp=sharing).
>
> > Q4: Additional iterations.
>
> We provide two more iterations and the results in the table below show that the performance converges. Due to the word limit, please see the results and discussion in our rebuttal to reviewer gyxq's Q1.
>
> |Win Rate|Iter. 1|Iter. 2|Iter. 3|Iter. 4|Iter. 5|
> |:-:|:-:|:-:|:-:|:-:|:-:|
> |Werewolf Side|0.54 (0.13)|0.63 (0.09)|0.73 (0.11)|0.75 (0.07)|0.76 (0.10)|
> |Villager Side|0.18 (0.09)|0.23 (0.12)|0.27 (0.11)|0.31 (0.09)|0.30 (0.07)|
>
> ---
> We greatly appreciate your thorough review of our work and hope our answers effectively address your concerns. We would be very grateful if you could consider raising the rating based on the responses.

---

### Official Review · Reviewer_gyxq · 2025-03-17

**Overall Recommendation:** 2

**Summary:**

The paper proposes an iterative framework called Latent Space Policy Optimization (LSPO) to develop strategic language agents for free-form social deduction games, using Werewolf as the testbed. The approach begins by mapping free-form discussion actions into a finite latent strategy space via embedding and k-means clustering, allowing the use of game-theoretic methods such as Counterfactual Regret Minimization (CFR) to learn near-optimal strategies. Subsequently, the learned latent policies are used to fine-tune a large language model (LLM) via Direct Preference Optimization (DPO), which in turn expands the latent strategy space in successive iterations. Experimental results, including latent space visualizations, prediction accuracy, win rates, and comparisons with baselines, show that the LSPO agent progressively improves its strategic performance in the Werewolf game.

**Claims And Evidence:**

The authors claim that the LSPO framework effectively mitigates intrinsic bias in free-form language action generation and enhances action coverage through iterative latent space expansion. The paper supports these claims with extensive experimental evidence in the Werewolf game, including:

- Visualizations that illustrate the evolution of latent strategy clusters over iterations.
- Quantitative metrics showing improvements in prediction accuracy and win rates.
- Comparisons with established baseline methods, where the LSPO agent achieves the highest performance metrics.
- Ablation studies that underscore the importance of LLM fine-tuning in the iterative process.

While the evidence is thorough within the context of the Werewolf game, it remains confined to a single domain, raising questions about generalizability.

**Essential References Not Discussed:**

-

**Experimental Designs Or Analyses:**

The experimental design is within the confines of the Werewolf game environment. The authors conduct self-play experiments, visualize the latent strategy evolution, and compare performance against multiple baselines. Ablation studies further validate the contribution of each component of the LSPO framework. Nonetheless, the experimental analysis is limited to one game setting, and the paper does not address the statistical significance or robustness of the observed improvements under varying hyperparameter choices.

**Methods And Evaluation Criteria:**

The methodology combines established techniques like embedding-based clustering, deep CFR for policy optimization, and DPO for LLM fine-tuning, into a cohesive, iterative framework. The paper presents a clear description of each component and details the experimental setup, including hyperparameters and evaluation metrics such as prediction accuracy and win rate.

**Other Comments Or Suggestions:**

-

**Other Strengths And Weaknesses:**

S:
- Innovative integration of latent space abstraction with iterative LLM fine-tuning
- Comprehensive experimental evaluation with detailed supplementary material
- Clear description of the methodology and comparisons with multiple baselines

W:
- Evaluation confined to a single game environment
- The iterative process, including clustering and fine-tuning, is complicated, may be computationally intensive and sensitive to hyperparameter settings.
- The contribution, while methodologically sound, seems incremental relative to existing methods, especially without a deeper theoretical analysis

**Questions For Authors:**

- Can you provide further insights or empirical evidence regarding the convergence behavior of the iterative LSPO process? Are there theoretical guarantees or observed trends that ensure the stability of the latent space over successive iterations?
- How sensitive is your method to the choice of clustering parameters (e.g., the number of clusters) and fine-tuning hyperparameters? Have you performed any sensitivity analyses to understand the robustness of your approach?
- Given that your evaluation is limited to the Werewolf game, do you have plans to test LSPO in other free-form language environments or strategic decision-making tasks? How do you anticipate your method will scale to different or more complex domains?
- The iterative process appears computationally intensive. Can you elaborate on the computational resources required and discuss any potential optimizations or trade-offs that could make the approach more scalable in practice?

**Relation To Broader Scientific Literature:**

The paper relates its contributions to prior works such as ReAct, ReCon, Cicero-like agents, and SLA, and draws upon established methods in CFR and reinforcement learning

**Theoretical Claims:**

The paper does not introduce new theoretical proofs but builds upon established methods like CFR and reinforcement learning.

---

> ### Author Rebuttal · Authors · 2025-04-01
>
> Thank you for your constructive comments and questions! We hope the following response can address your concerns.
>
> > Q1: Convergence behavior of the iterative LSPO process. (Also mentioned in Weakness 3)
>
> Theoretically, suppose the free-form language action has a finite vocabulary size $N_v$ and a finite maximum length $L$, then the language action space $N_v^L$ is also finite. Then with at most $N_v^L$ iterations, our method will cover the full language action space and the abstracted game becomes the original full game. And the LSPO process will converge in a finite number of iterations.
>
> Empirically, we observe that the iteration for convergence is much less than the theoretical upper bound. We perform two more iterations in the 7-player Werewolf game, and the results in the table below show that the performance converges.
>
> |    Win Rate   |   Iter. 1   |   Iter. 2   |   Iter. 3   |   Iter. 4   |   Iter. 5   |
> |:-----------:|:---------:|:---------:|:---------:|:---------:|:---------:|
> | Werewolf Side | 0.54 (0.13) | 0.63 (0.09) | 0.73 (0.11) | 0.75 (0.07) | 0.76 (0.10) |
> | Villager Side | 0.18 (0.09) | 0.23 (0.12) | 0.27 (0.11) | 0.31 (0.09) | 0.30 (0.07) |
>
> The early convergence compared to the theoretical upper bound is because the number of latent strategies in free-form language games is much smaller than the whole language space. For example, in the Werewolf game, the latent strategies for Werewolves can be roughly divided into pretending to be the Seer, pretending to be the Doctor, and pretending to be a Villager. More iterations lead to more fine-grained latent strategies, but the semantic meaning remains similar. Therefore, our method empirically converges with a few iterations.
>
> > Q2: Sensitivity and robustness analysis. (Also mentioned in Weakness 2)
>
> To perform sensitivity analysis on our method, we consider a simpler 4-player Werewolf game (1 Werewolf vs 1 Seer + 2 Villagers).
>
> We first perform experiments on the number of initial clusters $k=1, 2, 3$. The results in the table below show that the number of clusters can influence the performance in the early iterations, but does not influence the final performance.
>
> | Werewolf Win Rate |   Iter. 1   |   Iter. 2   |   Iter. 3   |
> |:---------------:|:---------:|:---------:|:---------:|
> |       $k=1$       | 0.13 (0.06) | 0.20 (0.10) | 0.24 (0.11) |
> |       $k=2$       | 0.21 (0.11) | 0.24 (0.12) | 0.25 (0.08) |
> |       $k=3$       | 0.23 (0.09) | 0.25 (0.06) | 0.25 (0.07) |
>
> We also perform ablations on the DPO hyperparameter $\beta=0.05, 0.1, 0.2$. The result in the table below shows that our method is robust to fine-tuning hyperparameters.
>
> | Werewolf Win Rate |   Iter. 1   |   Iter. 2   |   Iter. 3   |
> |:-----------------:|:-----------:|:-----------:|:-----------:|
> |    $\beta=0.05$   | 0.22 (0.10) | 0.24 (0.08) | 0.25 (0.08) |
> |    $\beta=0.1$    | 0.23 (0.11) | 0.24 (0.12) | 0.25 (0.08) |
> |    $\beta=0.2$    | 0.23 (0.07) | 0.25 (0.09) | 0.25 (0.06) |
>
> > Q3: Application to other games. (Also mentioned in Weakness 1)
>
> We perform experiments on two additional games to show the general applicability and effectiveness of our method. The two new games are Rock-Paper-Scissors-Spock-Lizard and Undercover (Who is Spy). Our method achieves the best performance in these two games. Due to the word limit of rebuttal, please see the detailed discussion in our rebuttal to reviewer sE2V's first concern.
>
> *Rock-Paper-Scissors-Spock-Lizard*
> |      Agent     | ReAct |  SLA | LSPO Iter. 1 | LSPO Iter. 2 | LSPO Iter. 3 |
> |:--------------:|:-----:|:----:|:------------:|:------------:|:------------:|
> | Exploitability |  0.42 | 0.33 |     0.33     |     0.33     |   **0.00**   |
>
> *Undercover*
> |   Agent  | ReAct | Cicero-like |  SLA |   LSPO   |
> |:--------:|:-----:|:-----------:|:----:|:--------:|
> |1 round|0.26|0.24|0.29|**0.32**|
> |2 rounds|0.15|0.16|0.20|**0.25**|
>
> > Q4: Computation requirement and trade-offs for scalability.
>
> Our method is trained on an 8xA800 GPU server with a 64-core CPU. The main computation bottleneck is fine-tuning the LLM (llama 3 8B) with DPO. Some trade-offs can be introduced to reduce the computation with some loss of performance.
> 1. Parameter-efficient fine-tuning: use techniques like LoRA (Low-Rank Adaptation) to update a small subset of parameters rather than the entire model and cut down the memory and computation usage.
> 2. Mixed precision training: use lower precision like fp16/bf16 to reduce memory consumption and accelerate training.
>
> Although these techniques can be used to further improve the scalability of our method, we would like to emphasize that the primary focus of our method is to address the intrinsic bias and action coverage issue of LLM agents in strategic games.
>
> ---
> We genuinely value your dedication to reviewing our paper and believe our detailed responses have addressed your concerns. We would really appreciate it if you could consider raising the rating of our work based on our responses.

---

### Official Review · Reviewer_sE2V · 2025-03-25

**Overall Recommendation:** 2

**Summary:**

This paper introduces Latent Space Policy Optimization (LSPO), an iterative framework designed to improve LLM-based agents in social deduction games like Werewolf. LSPO maps free-form text into a discrete latent space, enabling strategic policy learning using game-theoretic methods. It then translates the learned policy into natural language dialogues to fine-tune the LLM via DPO. Experimental results show LSPO improves agent performance and outperforms existing methods in Werewolf, demonstrating its potential for free-form language decision-making.

**Claims And Evidence:**

I don't think studying the performance on the Werewolf game alone with a specific criterion is sufficient to claim the effectiveness of the proposed method. A broader evaluation across different games or tasks with varying complexity could provide more convincing evidence of its general applicability and effectiveness.

**Essential References Not Discussed:**

No

**Experimental Designs Or Analyses:**

Yes

**Methods And Evaluation Criteria:**

My main concern is the evaluation criteria. Using win rate against other agents might not be sufficient since adversarial agents could be trained to win using unreasonable strategies. It would be more interesting to see if LLMs can discover human-like strategies or even novel strategies through their interactions.

**Other Comments Or Suggestions:**

No

**Other Strengths And Weaknesses:**

No

**Questions For Authors:**

When playing games with humans, humans can make misleading actions or claims. Since humans can use any random language to break LLM-based players, how would you tackle this issue?

**Relation To Broader Scientific Literature:**

No

**Theoretical Claims:**

No

---

> ### Author Rebuttal · Authors · 2025-04-01
>
> Thank you for your constructive comments and questions! We hope the following response can address your concerns.
>
> > Claims And Evidence: Evaluation on the Werewolf game alone is not sufficient.
>
> To show the general applicability and effectiveness of our method, we perform additional experiments on the following two games.
>
> *(1) Rock-Paper-Scissors-Spock-Lizard:* this is a [five-weapon variant](https://en.wikipedia.org/wiki/Rock_paper_scissors#Additional_weapons) of the classic Rock-Paper-Scissors game. Although it is not a free-form language game, we use it as a simple proof-of-concept game that highlights the motivations of our method: intrinsic bias in actions and inadequate action coverage of LLM-based agents.
>
> The Nash equilibrium (NE) in this game is to choose each action with an equal probability of 1/5. We use exploitability to evaluate the performance of different agents. A lower exploitability means the strategy is better (closer to NE).
>
> |Agent|ReAct|SLA|LSPO Iter. 1|LSPO Iter. 2|LSPO Iter. 3|
> |:-:|:-:|:-:|:-:|:-:|:-:|
> |Exploitability|0.42|0.33|0.33|0.33|**0.00**|
>
> Analysis:
> 1. ReAct fails because of the intrinsic bias issue. The LLM has higher probabilities to choose Rock, Paper, and Scissors, and much lower probabilities to choose Spock and Lizard, probably due to the bias inherited from pretraining data.
> 2. SLA fails because of the inadequate action coverage. SLA first uses LLM to propose $N$ actions and then uses RL to learn the optimal policy. A typical value is $N$=3 and results in a subgame without the other 2 actions. An NE in this subgame is not the NE of the original game.
> 3. Our method LSPO addresses these two issues by iteratively expanding the action space and using CFR to solve the NE. As the LSPO covers the full action space with 3 iterations, it learns the NE of the game.
>
> *(2) Undercover (Who is Spy):* this is a 5-player free-form language game introduced in LLMArena [1]. The players are divided into 2 groups: 1 undercover and 4 non-undercover. The two groups receive a pair of similar but distinct words (like apple and pear) and describe their words to find the undercover or hide.
>
> We consider 2 settings of this game: one lasts for 1 round, the other lasts for 2 rounds. We use the undercover's win rate against gpt-3.5-turbo in 100 games to evaluate the performance of different agents.
>
> |Agent|ReAct|Cicero-like|SLA|LSPO|
> |:-:|:-:|:-:|:-:|:-:|
> |1 round|0.26|0.24|0.29|**0.32**|
> |2 rounds|0.15|0.16|0.20|**0.25**|
>
> As shown in the table, our method achieves the highest win rates in both settings. This is because the LSPO agent learns the action of deception, i.e., realizes itself is an undercover and follows the descriptions of other agents. This helps the LSPO agent to blend in with the non-undercovers and achieves a higher win rate.
>
> [1] Chen, Junzhe, et al. "LLMArena: Assessing Capabilities of Large Language Models in Dynamic Multi-Agent Environments." ACL (1). 2024.
>
> > Method And Evaluation Criteria: Using win rate alone is not sufficient, it would be more interesting to see human-like or even novel strategies.
>
> We would like to clarify that, besides win rates, we also use visualization with discussion of latent policies for qualitative evaluation (Sec. 4.1) and prediction accuracy for quantitative evaluation (Sec. 4.2) in our manuscript.
>
> As discussed in Sec. 4.1, as training proceeds, our method discovers emergent human-like behaviors. For the Werewolves, our method learns to pretend to be the Seer and provide fabricated investigations to sow confusion. For the Seer, our method learns a coordination voting strategy that asks all players to vote for a strongly suspected Werewolf. Due to the word limit of rebuttal, please see more detailed game log in the [rebuttal supplementary](https://drive.google.com/file/d/1Ft_GLmeIwLdFR246ApV0b_vE4qb04OOU/view?usp=sharing).
>
> In addition, the prediction accuracy result in Table 1 also shows that our agent is not winning with "unreasonable strategies". Instead, it successfully predicts the hidden roles of other players and wins by strong strategic play.
>
> > Questions For Authors: When playing games with humans, humans can make misleading actions or claims. Since humans can use any random language to break LLM-based players, how would you tackle this issue?
>
> Since our method does not restrict the agents' actions to the latent strategies, the LLM agents can also use any random language to exploit their opponents like humans. To make them more robust, we train our agents using self-play with CFR, which has a theoretical regret bound. We also provide some examples in the [rebuttal supplementary](https://drive.google.com/file/d/1Ft_GLmeIwLdFR246ApV0b_vE4qb04OOU/view?usp=sharing) to show that our agents stay robust to deliberate exploitation from humans.
>
> ---
> We genuinely value your dedication to reviewing our paper. We have carefully addressed each of your concerns, and we sincerely hope our efforts merit a raise in your rating.

---

### Decision · Program_Chairs · 2025-05-01

**Decision:**

Accept (poster)

**Comment:**

This paper introduces Latent Space Policy Optimization (LSPO), a novel framework for training LLM-based agents in social deduction games like Werewolf, which require both strategic reasoning and natural language interaction. By mapping free-form text into a discrete latent space for policy learning and iteratively refining the agent through Direct Preference Optimization (DPO), LSPO significantly improves performance over existing agents in both decision-making and dialogue quality.

I find this line of work, integrating language-based communication and decision-making, to be very important. While the reviewers raised several points, I believe the repeated concern about evaluating only a single setup is relatively weak, as implementing such methods even for one game is far from trivial. I appreciate the reviewers’ additional comments, and as an area chair, I face  the challenge of weighing the pros and cons. All in all, the concerns do not seem particularly significant to me, and the paper’s novelty and potential impact are substantial.

Previous work: I recommend the authors to look on recent papers by Eilam Shapira et al., which address similar setups.